# On Optimal Hyperparameters for Differentially Private Deep Transfer Learning

**Aki Rehn[1*], Linzh Zhao[1*], Mikko A. Heikkilä[1], Antti Honkela[1]**

[1]Department of Computer Science, University of Helsinki, Finland

[*]Equal contribution

{aki.rehn, linzh.zhao, mikko.a.heikkila, antti.honkela}@helsinki.fi

## Abstract

Differentially private (DP) transfer learning, i.e., fine-tuning a pretrained model on private data, is the current state-of-the-art approach for training large models under privacy constraints. We focus on two key hyperparameters in this setting: the clipping bound $C$ and batch size $B$. We show a clear mismatch between the current theoretical understanding of how to choose an optimal $C$ (stronger privacy requires smaller $C$) and empirical outcomes (larger $C$ performs better under strong privacy), caused by changes in the gradient distributions. Assuming a limited compute budget (fixed epochs), we demonstrate that the existing heuristics for tuning $B$ do not work, while cumulative DP noise better explains whether smaller or larger batches perform better. We also highlight how the common practice of using a single $(C, B)$ setting across tasks can lead to suboptimal performance. We find that performance drops especially when moving between loose and tight privacy and between plentiful and limited compute, which we explain by analyzing clipping as a form of gradient re-weighting and examining cumulative DP noise.

## 1    Introduction

The current state-of-the-art approach for training large models on sensitive data is differentially private (DP) *transfer learning*: after pretraining a backbone on non-private data, the model is fine-tuned on the private task using a DP optimizer, such as DP-SGD or DP-Adam (De et al., 2022). Due to the high computational requirements of DP optimization, commonly only the learning rate is tuned for each separate problem, while other hyperparameters, most importantly batch size and/or clipping bound, are assumed to be stable and fixed to a single value across different privacy levels, backbones, and computational budgets (see, e.g., De et al. 2022; Panda et al. 2024; Sander et al. 2023).

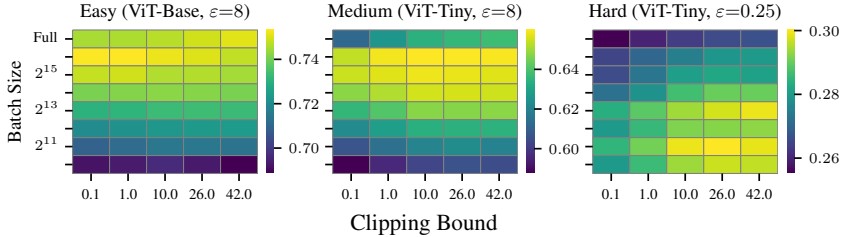

Figure 1: Macro accuracy heatmaps under increasing learning-problem difficulty (left to right): strong model with loose privacy, weaker model with same privacy, and weaker model with tight privacy. Experiments use DP-Adam with $\delta = 10^{-5}$ on SUN397 ('Full'=87002), averaged over 5 seeds with 8 epochs each. Learning rates are tuned separately for each configuration using a fixed grid. No single clipping bound or batch size performs best across all settings.

In this paper, we demonstrate that fixing hyperparameters often leads to suboptimal performance and explain why this happens. We find across multiple datasets that the optimal clipping bound and batch size are affected by the overall *learning–problem difficulty*—that is mainly governed by the privacy budget, available data and compute, dataset difficulty, transfer complexity (Tobaben et al., 2023),

and the pretrained backbone capability, which reflects both model complexity (Hu et al., 2021b) and pretraining quality.

Figure 1 illustrates how fixing the clipping bound and batch size across problems fails to account for this difficulty: the settings that perform best in easy cases degrade clearly under harder ones, and vice versa. Ignoring these shifts and fixing clipping bound and batch size across tasks systematically suppresses DP transfer learning performance, reducing overall accuracy and especially harming difficult examples and the classes they dominate. We summarize implications for hyperparameter tuning in Table 1.

Table 1: Practical implications for hyperparameter tuning.

| Change in condition | Tuning implication |
|---|---|
| Lower $\varepsilon$ (tighter privacy) | Increase $C$, decrease $B$. |
| Stronger backbone / easier dataset | Use smaller $C$. |
| Weaker backbone / harder dataset | Try larger $C$. |
| Fewer epochs (limited compute) | Avoid large $B$ ($\Rightarrow$ more steps). |
| More epochs | Larger $B$ becomes viable. |
| General guidance | Tune $(C, B, \eta)$ jointly for each task. |

**Contributions**

1. We present a systematic study of how the optimal clipping bound $C$ and batch size $B$ vary—primarily with the privacy budget ($\varepsilon$), but also with other factors that affect the difficulty of the private fine-tuning task, such as the capability of the pretrained backbone and the amount of compute—in *DP deep transfer learning*.
2. We demonstrate through extensive experiments that contrary to what previous theory suggests, increasing $C$ can improve results with smaller $\varepsilon$ (Section 5.1), and with less capable pretrained models (Section 5.2). To explain these findings, we derive a novel optimal clipping result (Theorem 5.2) and show how it affects optimization progress (Theorem 5.4, Corollary 5.5), which results might be of independent interest.
3. We propose a rule for selecting the optimal batch size under bounded compute (fixed epochs) through a combination of a lower bound on the number of optimisation steps and maximising the number of steps without increasing the total cumulative DP noise (Section 6).

## 2 BACKGROUND

DP (Dwork et al., 2006b) provides a formal guarantee that an algorithm's output distribution does not depend too strongly on any single individual's record. This guarantee is defined with respect to *neighboring datasets*, which differ by one record. Formally:

**Definition 2.1.** *(Approximate differential privacy, Dwork et al., 2006b;a)*

*An algorithm $\mathcal{M} : \mathcal{D} \to \mathcal{R}$ is $(\varepsilon, \delta)$-differentially private if for all neighboring datasets $D, D' \in \mathcal{D}$ and for all subsets $S \subseteq \mathcal{R}$, it holds that*

$$\Pr[\mathcal{M}(D) \in S] \leq e^{\varepsilon} \Pr[\mathcal{M}(D') \in S] + \delta.$$

We use the *add-remove* neighborhood relation with *sample-level privacy*, meaning the neighboring datasets differ by the presence or absence of a single training example. $\varepsilon > 0$ and $\delta \in [0, 1]$ control the strength of the privacy guarantee with smaller values implying stronger privacy. We set $\delta = 10^{-5}$ in all the experiments.

In machine learning, DP is typically enforced by adjusting the optimization process by suitably randomizing each optimizer step. Algorithms such as DP-SGD (Song et al., 2013; Abadi et al., 2016; Rajkumar & Agarwal, 2012) and DP-Adam (Abadi et al., 2016; Kingma & Ba, 2015) clip per-example gradients to a fixed norm in order to bound the *sensitivity*—the maximum change any single record can have on the model update. The optimizer then adds noise, calibrated to the sensitivity, to the aggregated gradients to guarantee DP. Each training iteration then contributes to the cumulative privacy loss. We use the Privacy Random Variable (PRV) accountant (Gopi et al., 2021), a numerical method that tightly tracks the privacy guarantees under subsampling, to calibrate the noise level to match the target privacy budget.

Gradient clipping and noise addition generalize to any gradient-based optimizer. We focus on DP-Adam due to its widespread use, but the same principles apply to any first-order optimization method (McMahan et al., 2019). In experiments, we use a variant that decouples the learning rate and the clipping bound (Algorithm 1). For analysis we adopt the standard (unnormalized) variant in which the DP noise term scales explicitly with $C$. See Appendix B for details of these two variants.

## 3 RELATED WORK

In this work, we mainly focus on analyzing two hyperparameters in DP optimization: clipping bound $C$ and batch size $B$. As these are critical parameters that need to be somehow addressed by every DP deep learning work, we limit our discussion to the most significant works.

**Clipping bound** There are some theoretical reasons to expect that the clipping bound $C$ is an important hyperparameter for model training: Chen et al. (2020) and Koloskova et al. (2023) show that clipping in DP-SGD will in most cases cause bias. The bias due to clipping tends to amplify fairness disparities (see, e.g., Bagdasaryan et al. 2019; Esipova et al. 2023).

---

**Algorithm 1** Generic DP optimization, normalized, from (De et al., 2022)

---

**Input:** iterations $T$, dataset $D$, sampling rate $q$, clipping bound $C$, noise multiplier $\sigma$, learning rate $\eta$, initial weights $\theta_0$, initial optimizer state $\mathcal{O}_0$.
**for** $t = 0, \ldots, T - 1$ **do**
    $B \sim \text{PoissonSample}(D, q)$
    **for** $(x_i, y_i) \in B$ **do**
        $g_i \leftarrow \nabla \mathcal{L}(f_{\theta_t}(x_i), y_i)$
        $g_i^{\text{clip}} \leftarrow g_i \cdot \min\left(\frac{1}{C}, \frac{1}{\|g_i\|_2}\right)$
    **end for**
    $\bar{g} \leftarrow \frac{1}{|B|}\left(\sum_{i \in B} g_i^{\text{clip}} + \mathcal{N}(0, \sigma^2 \mathbf{I}_d)\right)$
    $(\theta_{t+1}, \mathcal{O}_{t+1}) \leftarrow \text{OptimizerStep}(\theta_t, \bar{g}, \eta, \mathcal{O}_t)$
**end for**

---

For properly tuning a fixed $C$ in practice, Ponomareva et al. (2023) propose using a non-private model to identify the smallest $C$ that slightly degrades utility, then applying this during private training, while Tobaben et al. (2023) perform Bayesian optimization over $C \in [0.2, 10]$ to directly optimize utility.

However, many recent papers simply fix $C$ to a constant for all experiments, e.g., to $C = 0.1$ (Tramèr & Boneh, 2021) or $C = 1$ (De et al., 2022; Berrada et al., 2023; Sander et al., 2023; Panda et al., 2024; Koskela & Kulkarni, 2023). In the same vein, Bu et al. (2023) eliminate the need to set any specific $C$ by introducing automatic clipping, which normalizes each per-sample gradient individually, which is equivalent to setting an extremely small clipping bound. Liu & Bu (2024) further build on this idea and propose a hyperparameter-free framework that combines automatic clipping with adaptive learning rate schedules.

Adaptive clipping (Andrew et al., 2021) translates the problem of choosing a fixed $C$ to estimating it dynamically from a quantile of the gradient norm distribution. The adaptive approach can still lead to suboptimal results for example with highly bimodal gradient norm distributions that commonly arise under strong privacy. As a final alternative, Zhang et al. (2024) introduced an error-feedback method to trade the elimination of clipping bias to adding more noise.

In summary, there is plenty of contrasting and partially contradicting claims and recommendations on the importance and tuning of clipping bound in the existing literature. In this work we seek to clarify these issues.

**Batch size** Our work focuses on the bounded compute setting, which we implement through a bound on training epochs. This is in contrast to prior work mostly assuming a fixed number of training steps. The fixed-epochs setting introduces a trade-off between batch size and the number of steps.

Based on theoretical analysis of a single-step setting, Räisä et al. (2024) show that effective DP noise variance, $\sigma^2/q^2$, where $\sigma$ is DP noise and $q$ is the subsampling probability, monotonically decreases as batch size increases. As the number of fixed steps increases, the effective DP noise variance becomes invariant of $B$, leaving only the mini-batch induced variance which decreases as $B$ increases. Based on similar reasoning and the fixed-steps setting, Ponomareva et al. (2023) recommend large batches, but note a point of "diminishing returns", beyond which performance plateaus.

Empirically, Abadi et al. (2016) note that batch size has a relatively large impact on accuracy and suggest approximately $\sqrt{N}$ as the optimal batch size for a dataset with $N$ observations. Conversely, Panda et al. (2024) find that under their linear learning rate scaling, batch size has only very small impact.

Recent empirical work in DP deep learning generally recommend using relatively large batch sizes. In image classification, De et al. (2022) show that without any other constraints, increasing the batch size monotonically increases accuracy over a broad range of values, but larger batches require more epochs and thus more compute. Mehta et al. (2023) show that for certain models and tasks, performance may improve up to batch sizes of 1M.

In large language models, McKenna et al. (2025) study the optimal batch size by simulating the effect of varied batch sizes from experiments performed using fixed physical batch size of 1024. For largest compute budgets, the optimal simulated batch size can exceed 1M. They also make the surprising discovery that even smaller physical batches can perform better in high noise (low $\varepsilon$) settings.

Some works treat batch size as a fixed, data-independent hyperparameter. Bu et al. (2023); Liu & Bu (2024) argue that batch size does not need tuning, while Morsbach et al. (2024) consider it unimportant. In few-shot settings, Tobaben et al. (2023) include a wide range of batch sizes (including full-batch) in their hyperparameter search, though they do not analyze its effect.

While much of the recent work favors large batches, we find that under fixed-epoch settings—typically imposed by limited compute—moderate-sized batches (see Section 6) can outperform large ones, especially under high learning-problem difficulty, where more iterations are needed for convergence.

## 4 METHODOLOGY

**Models** For image tasks, we employ Vision Transformers (ViTs) (Dosovitskiy et al., 2021) from the PyTorch Image Models library (Wightman, 2019), using ViT-Base and ViT-Tiny to represent high- and low-capability pretrained backbones, respectively. All image models for fine-tuning tasks are pretrained on ImageNet-21k (Ridnik et al., 2021) with the "AugReg" setup that includes multiple data augmentation methods as well as regularization methods (Steiner et al., 2022). Depending on the classifier head size, ViT-Base has approximately 85M parameters, and ViT-Tiny has around 5M. For text classification, we use the DistilBERT Sanh et al. (2019) model which has roughly 66M parameters. Moreover, the experiments for training from scratch employs WideResNet-16-4 (Zagoruyko & Komodakis, 2016), which contains about 2.8M parameters.

**Fine-tuning & from-scratch training** For image tasks, we follow the experimental setup of Tobaben et al. (2023) for differentially private fine-tuning. Specifically, we use the feature-wise linear modulation (FiLM) parameterization (Perez et al., 2018) by freezing all the other parameters and train only the scale and bias of the normalization layers in addition to the classification head. This results in approximately 0.5–1.5% of trainable parameters, depending on the size of the backbone and the classification head. Following the DP-FiLM methodology, we initialize the classification head weights to zero. Furthermore, we also experiment training with low-rank adaptation (LoRA) (Hu et al., 2021a) under DP with image models (see Appendix G.1), and lastly with with full fine-tune (see Appendix I.3). For natural language tasks, we follow the standard practice of fine-tuning them using LoRA. For training from scratch, we use the WideResNet-16-4 (Zagoruyko & Komodakis, 2016), also used in from scratch experiments by De et al. (2022), a CNN with approximately 2.7M parameters, on the CIFAR-10 dataset.

**Hyperparameter grids** We perform a structured grid search over all three hyperparameters: learning rate, batch size, and clipping bound (see Table A1). Batch sizes range exponentially from 256 up to full-batch (i.e., the full training); the range of clipping bounds depend on the dataset; and similarly learning rates are chosen geometrically from dataset-specific ranges. We provide details in Appendix C.

We found this exhaustive tuning is necessary to expose the subtle interactions between DP-specific hyperparameters and training dynamics. For instance, the learning rate often dominates optimization, masking the influence of the other hyperparameters in less thorough hyperparameter tuning methods.

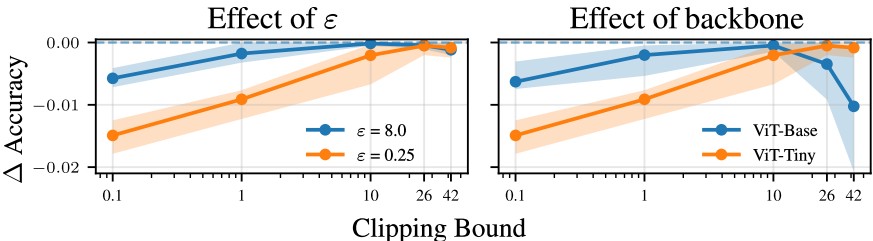

Figure 2: Accuracy difference to the best, mean over 5 repeats with min–max bands; SUN397 dataset, 8 epochs, $\delta=10^{-5}$. Learning rate and batch size are tuned jointly with clipping bound for each case. Left: optimal clipping constant is larger with tighter privacy (small $\varepsilon$). Right: low-capability pretrained backbone (ViT-Tiny) performs better with larger clipping bounds compared to a high-capability model (ViT-Base); $\varepsilon=0.25$. In both cases, harder tasks (orange) shift the optimal clipping bound upward, while easier tasks (blue) tolerate or even benefit from smaller bounds.

**Datasets** We evaluate models on four datasets: SUN397 (Xiao et al., 2016; 2010), Cassava (Mwebaze et al., 2019), CIFAR-100 (Krizhevsky & Hinton, 2009), and 20 Newsgroups (Mitchell, 1997) (see Appendix C.2 for details). Additionally, we include a 10% subset of CIFAR-100 to (i) study how scarcity of training data increases learning-problem difficulty, and (ii) analyze how models capable of overfitting the training dataset affect the clipping bound. For each dataset, we merge the validation split (if available) into the training set to maximize training data for each configuration. We evaluate the final accuracy on the test dataset. SUN397 is a highly imbalanced real-world scene recognition dataset with 397 classes and $\sim 87$k training examples. Cassava is an imbalanced plant disease classification dataset with 5 classes and $\sim 6$k training examples, and is significantly out-of-distribution relative to the ImageNet-21k pretraining dataset. CIFAR-100 is widely used, balanced natural image classification dataset with 100 classes and 50k training examples. 20 Newsgroups is a text classification dataset consisting of roughly 18k posts taken from 20 online newsgroups. For the image datasets, all input images are resized to $224 \times 224$ resolution to match the pretraining dimensionality.

## 5 OPTIMAL CLIPPING DEPENDS ON PROBLEM DIFFICULTY

In this section, we argue that the optimal clipping bound depends fundamentally on the learning problem difficulty, and consider clipping as a form of gradient re-weighting. Fig. 2 shows results for ViT-Tiny on SUN397 under two different privacy levels (left) and for two pre-trained backbones (right) for varying $C$: with easy problem (shown in blue; left larger $\varepsilon$, right: more capable backbone ViT-Base) optimal $C$ is clearly smaller than with a hard problem (shown in orange; left: small $\varepsilon$, right: less capable backbone ViT-Tiny). [1] We provide additional image results (including ResNet-50) in Appendix F, text classification results in Appendix G.2, results with LoRA fine-tuning in Appendix G.1, and lastly from scratch and DP-SGD training results in Appendix I.1, all of which appear to show connection between clipping bound and learning-problem difficulty. Lastly, the odd-one out is fine-tuning all the model parameters , where we find that—especially with large models—the increased parameter count, resulting in larger average gradient norms—prevents using smaller clipping bounds and rendering both easy and hard task to prefer large $C$ (see detailed experiments and discussion in Appendix I.3).

In the rest of this section, we first analyze how these two factors affecting problem difficulty, DP noise level and pretrained model capability, affect the optimal clipping bound, in Section 5.1 and Section 5.2, respectively. To understand better how clipping affects the gradient distributions, we look at clipping as a form of gradient re-weighting in Section 5.3.

Considering hyperparameter tuning, our results imply that using any fixed $C$ should give close to optimal performance only for settings of similar difficulty, and especially that using a small $C$ only works with easy-enough settings.

---

[1]Note that we do not claim that for decreasing $\varepsilon$ a smaller $C$ is *always* better, rather, we want to understand when and why this is the case. See Appendix F for more results.

## 5.1 How DP noise affects optimal clipping

De et al. (2022) note that gradient clipping introduces a bias–variance tradeoff in DP optimization, but do not provide a formal analysis. The closest theoretical work connecting DP-SGD optimization performance to the DP noise level $\sigma$ and the clipping bound $C$ comes from Koloskova et al. (2023), who provide convergence guarantees for DP-SGD with per-example gradient clipping. However, any optimal clipping bound derived from their work would depend on quantities that are not known in advance (such as the loss evaluated at the unknown optimum; see Appendix D). Furthermore, these bounds do not explain why—as seen in our experiments—tighter privacy can favor larger clipping bounds.

To understand when and why this happens, we analyze clipping in DP-SGD through a mean squared error (MSE) decomposition into clipping bias and DP noise variance. This allows us to characterize how the tradeoff depends on both the noise level and the gradient norm distribution, and to derive a clipping bound that minimizes the per-step gradient MSE. We further connect this MSE to optimization progress, showing that reducing this MSE tightens the bound on per-step optimization progress.

To understand the empirical results, we therefore derive the optimal clipping threshold under a standard unnormalized DP-SGD formulation. In this setting, per-example gradients are clipped individually and isotropic Gaussian noise with per-coordinate variance $\sigma^2 C^2$ is added. In the following, we use the standard (unnormalized) DP optimizer variant, where gradients are clipped individually and isotropic Gaussian noise with per-coordinate variance $\sigma^2 C^2$ is added. This variant adds the same amount of DP noise as the normalized one used in our experiments (see Appendix B).

**Assumption 5.1.** *Assume there is no mini-batch sampling noise, we use standard per-sample constant clipping with $C \in [\min_i \|g_i\|, \max_i \|g_i\|]$, and the Gaussian mechanism with noise level $\sigma$ to privatize the sum of clipped gradients.*

**Theorem 5.2.** *Under Assumption 5.1, an optimal clipping constant $C^*$ that minimizes the mean squared error between the per-sample clipped DP gradient $\tilde{g}$ and the true gradient $g$ for a fixed mini-batch satisfies*

$$C^* = \begin{cases} \|g_i\| \text{ for some } i, \text{ or} \\ \frac{N_{C^*}^T G_{C^*}}{N_{C^*}^T N_{C^*} + \sigma^2 d}, \end{cases} \tag{1}$$

*where $d$ is the dimensionality of the gradient, $G_C := \sum_{i \in I_C} g_i$ and $N_C := \sum_{i \in I_C} \frac{g_i}{\|g_i\|}$, and $I_C = \{i : \|g_i\| > C\}$ denotes the indices of the clipped gradients.*

*Proof.* See Appendix E for a proof. □

Note that we do not use Theorem 5.2 to find $C^*$ for doing DP optimization (this would require additional DP mechanisms as both Assumption 5.1 and Theorem 5.2 depend on the actual gradients), but only to describe how the clipping works. Next, we connect the per-step gradient MSE to the optimization performance. We again first state the assumptions and then give the theorem.

**Assumption 5.3.** *Assume an L-smooth loss function $\mathcal{L}$ (Definition E.1), and that we want to minimize $\mathcal{L}$ via gradient descent, using step size $\eta \leq \frac{1}{L}$, and the per-sample clipped and noisy DP gradients $\tilde{g}_t = \sum_i \bar{g}_i^{(t)} + \xi^{(t)}$ at step $t = 1, \ldots, T$ instead of the true gradients $g_t = \sum_i g_i^{(t)} = \nabla \mathcal{L}(\theta_t)$.*

**Theorem 5.4.** *Under Assumption 5.3, the expected improvement in the loss from step $t$ to $t+1$ is bounded as*

$$\mathbb{E}[\mathcal{L}(\theta_{t+1})|\theta_t] \leq \mathcal{L}(\theta_t) - \frac{\eta}{2}\|\nabla\mathcal{L}(\theta_t)\|_2^2 + \frac{\eta}{2}\text{MSE}_t(C) \tag{2}$$

*Proof.* See Appendix E for a proof. □

The following Corollary 5.5 follows immediately from Theorem 5.4, as $\text{MSE}(C)$ is non-negative.

**Corollary 5.5.** *Under Assumption 5.3, minimizing $\text{MSE}(C)$ minimizes the upper bound given in Theorem 5.4 for the expected per-step improvement in the loss function at any given step $t$.*

In summary, from Theorem 5.2 we see that $C^*$ depends not only directly on $\sigma$, but also on the actual gradients: as a simple example, assuming that the optimal clipping is not exactly one of the gradient magnitudes as well as constant gradient direction and fixed number of clipped gradients, then increasing $\sigma$ implies smaller $C^*$ if the gradient distribution is not much affected by the noise, while larger gradient norms would always push towards larger $C^*$. We can also connect $C$ directly to the actual optimization performance: as we show in Corollary 5.5, under Assumption 5.3 using optimal clipping that minimizes MSE directly improves the bound on the per-step loss improvement.

As shown in Fig. 3, the true gradient norm distributions do shift toward larger gradient norms under tighter privacy; in Appendix L we empirically verify that the gradients shrink toward the end of training. Appendix J shows the similar plots for $C = 0.1$ and $C = 42$. We argue that this more complex picture explains the empirical results we see with real data (Fig. 2, left), where smaller $\varepsilon$ can counterintuitively benefit from a larger clipping bound: increasing $\sigma$ shifts the gradient distributions, effectively pushing the terms in Eq. (1) towards larger values as the examples that were easy to learn with larger $\varepsilon$ become harder.

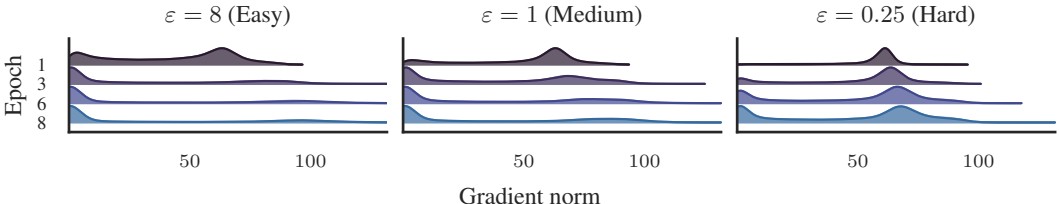

Figure 3: Gradient norm distributions over training epochs for ViT-Tiny on SUN397 with FiLM; 8 epochs, $\delta=10^{-5}$, $C=1$). Learning-problem difficulty increases from left to right as $\varepsilon$ decreases. Thicker regions imply higher probability mass. As difficulty increases, the distributions shift toward larger gradient norms.

## 5.2 HOW PRETRAINED MODEL CAPABILITY AFFECTS OPTIMAL CLIPPING

As mentioned in Section 5, Fig. 2 shows how the choice of pretrained model can have similar effect on the optimal clipping bound as changing $\varepsilon$ (switching to a more capable backbone can roughly correspond to using larger $\varepsilon$ for a fixed backbone). Furthermore, Theorem 5.2 explicitly shows how the optimal clipping constant $C^*$ depends on the DP noise $\sigma$, the subsampling rate $q$ (batch size), and the gradient distribution.

Considering how $C^*$ in Theorem 5.2 is affected by switching the pretrained backbone from a less capable one to a more capable one under fixed $\sigma$, as previously hard examples are now easier to classify, this generally reduces the gradient norms. For example, in the simple case where the gradient directions or the number of clipped gradients does not change, this directly leads to smaller $C^*$ in Theorem 5.2. While the true effect of changing the backbone is obviously more complex, as the gradient directions and number of clipped gradients can also be affected, we argue that this general effect the backbone has on the gradient distributions explains our empirical results.

Others have explored the features and their effect on learning: Tramèr & Boneh (2021) discussed this prior the DP fine-tuning era. More recently, Wang et al. (2024) find that high-quality features make learning more robust; aligning with our findings. Lastly, Zhao et al. (2025) note that poorly tuned hyperparameters (especially the learning rate) degrade the feature quality of the pretrained model.

## 5.3 CLIPPING AS GRADIENT RE-WEIGHTING

To understand how clipping interacts with shifts in the gradient distribution on a more granular level, besides just possibly causing bias (see, e.g., De et al. 2022; Koloskova et al. 2023), we can interpret clipping as a form of gradient re-weighting: decreasing $C$ gives more weight to easy examples/classes while down-weighting the harder ones, whereas larger $C$ weights all examples/classes more equally. We formalize this by defining the *retained weight* for class $y$ at clipping $C$ as

$$w_y(C) = \frac{1}{n_y} \sum_{i:y_i=y} \min(1, \frac{C}{\|g_i\|_2}), \qquad (3)$$

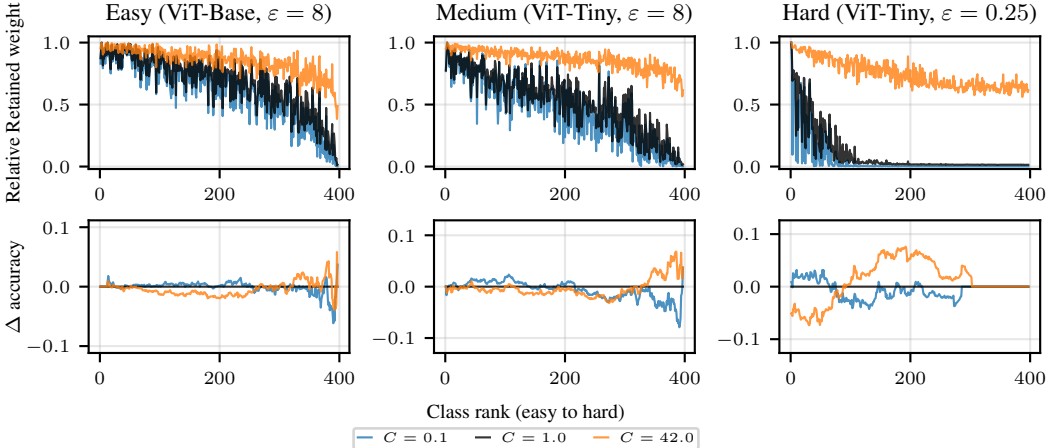

Figure 4: Per-class effects of gradient clipping under increasing task difficulty (left to right), SUN397, 8 epochs; $\delta{=}10^{-5}$. Class rank is based on sorting classes according to their per-class accuracies. Top: relative retained weight after clipping, computed per class and normalized to the class with the largest retained weight in the baseline ($C{=}1$). Flatter and higher curves indicate that gradients are preserved more uniformly across classes, while steep drops reveal that hard classes suffer disproportionately from clipping. Small clipping bounds risk over-clipping gradients from hard classes, while large bounds better preserve gradient signal across the classes under high learning-problem difficulty.

where $n_y = \sum_i \mathbb{I}(y_i = y)$ (number of samples with label $y$). Retained weight values near 1 mean gradients are mostly preserved; lower values indicate stronger clipping. This is shown in Fig. 4, where the top panel shows per-class retained weight (see Appendix C.4 for details; see Appendix K for the experiment settings, Appendix G.2 for results on text classification, Appendix I.1 for the training from scratch on image classification, and Appendix I.2 for fine-tuning with DP-SGD) and the bottom panel accuracy relative to the $C{=}1$ baseline, with learning-problem difficulty increases from left to right. As difficulty increases, the gap between clipping bounds widens; with small $C$ hard classes are severely down-weighted compared to easier ones. A larger $C$ better preserves hard-class gradients, recovering their performance at the cost of slightly reduced accuracy on easier classes. In short, clipping acts as a tunable knob that re-weights example (and class) contributions—its impact grows more asymmetric as learning-problem difficulty increases.

This re-weighting view to clipping explains why methods for tuning $C$ based on optimizing the performance without DP noise to save compute (e.g., Ponomareva et al. 2023) only work well in specific settings: optimizing $C$ without DP noise works in the regime where adding noise does not make the problem much harder (in the sense of not affecting the gradient distributions too much, cf. Fig. 3). Similarly, we would expect automatic clipping (Bu et al. 2023, essentially, using a very small $C$; see Appendix H) to work well in cases where focusing overwhelmingly on the easiest examples suffices for good performance or when class distributions are balanced so that no single group dominates the training. As shown in Fig. 5, this matches our empirical results: automatic clipping roughly matches tuned $C$ in easy-enough settings (larger $\varepsilon$, simpler dataset), while it clearly under-performs under harder settings (smaller $\varepsilon$, more complex dataset).

## 6 OPTIMAL BATCH SIZE DEPENDS ON PRIVACY AND COMPUTE BUDGETS

We study how batch size affects model performance under limited compute, focusing on the fixed-epochs setting. The number of epochs explains the compute quite well because the computational cost is dominated by feedforward and feedback computations that scale with the total number of samples processed. Under this setting, doubling the batch size halves the number of optimizer steps.

Prior works (Ponomareva et al., 2023; Räisä et al., 2024) take a fixed-steps view and suggest tuning the batch size by finding the sweet spot when plotting the standard deviation of the DP noise in the averaged gradient, thereby finding the smallest batch size with (nearly) optimal per-step signal-to-noise ratio.

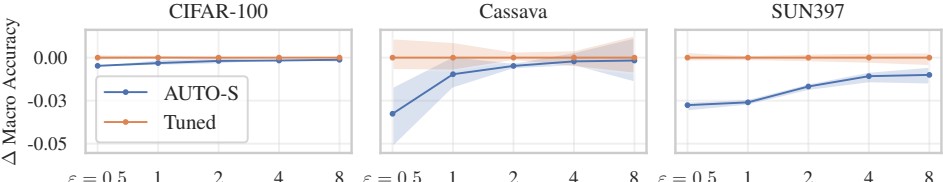

Figure 5: Comparison of `AUTO-S` (Bu et al., 2023) and properly tuned flat clipping, mean accuracy with min–max bands over 3 repeats, ViT-Tiny model, 8 epochs training for SUN397 and CIFAR-100, 32 epochs for Cassava; $\delta = 10^{-5}$. Lines show difference to best accuracy. Batch size and learning rate were tuned over a fixed grid for both methods. `AUTO-S` performs notably worse on harder datasets (SUN397, Cassava), especially under tight privacy, as predicted by our analysis. See Appendix H for absolute accuracies.

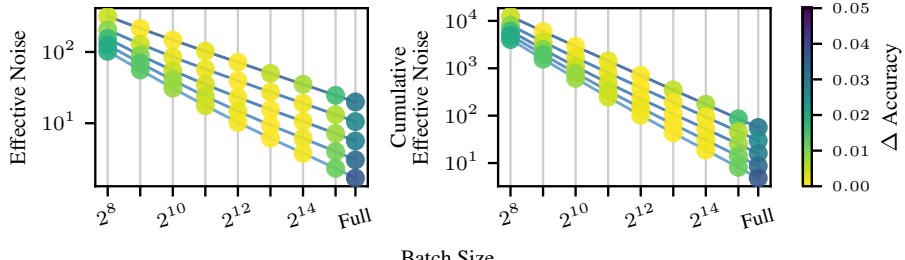

Figure 6: Neither per-step averaged gradient noise standard deviation suggested by Ponomareva et al. (2023) (left) or its cumulative version (right) saturates in the fixed-epochs setting, thereby failing to provide useful guidance for selecting the optimal batch size. The lines show the noise levels needed for fine-tuning ViT-Tiny for CIFAR-100 for 8 epochs with $\varepsilon = 0.5, 1, 2, 4, 8$ (from top to bottom), $\delta = 10^{-5}$. The color indicates the difference to best accuracy at each privacy budget; brighter is better.

The same algorithm for finding the optimal batch size does not work in the fixed-epochs setting. As shown in Fig. 6, the standard deviation plot does not have a sweet spot but would always suggest full batch which is suboptimal.

To develop an alternative that is better-suited to the fixed-epochs setting, we analyze how DP noise accumulates when the number of steps $T = E \cdot {}^{N}/{}_{B}$ increases as the batch size decreases. We compute the noise multiplier, $\sigma$, using the PRV accountant (Gopi et al., 2021) and from that derive the *cumulative noise* standard deviation: $\sigma\sqrt{T}$. This cumulative noise captures the total DP noise accumulated over training steps. We find that the cumulative noise, under the constraint of a minimum number of steps, can explain which batch sizes perform the best.

As Fig. 7 depicts, using the cumulative DP noise, we recover a weaker form of the plateau that is reported for average gradient noise standard deviation: in low-$\varepsilon$ regimes, $\sigma\sqrt{T}$ remains nearly constant across a wide range of batch sizes.

In addition to the cumulative noise, the number of steps plays an important role in determining the optimal batch size. This is illustrated in the right panel of Fig. 7 which shows that a minimum number of steps (indicated by the vertical dashed line at 20 steps in this case) is needed for good performance. As illustrated for SUN397 in Fig. A35, the actual minimum number can vary between datasets. Increasing the number of epochs pushes training further into the asymptotic regime, flattening the $\sigma\sqrt{T}$ curves and reducing sensitivity to batch size, similarly as in the asymptotic theory of Räisä et al. (2024).

As a rule, tighter privacy (lower $\varepsilon$) broadens the flat region in the cumulative noise and favors smaller batches compared to weaker privacy. The effect of increasing epochs is more subtle: it simultaneously allows using larger batches while still meeting the minimum steps requirement as well as broadens the flat region in cumulative noise.

# 7 DISCUSSION & CONCLUSIONS

We have presented a systematic study of factors influencing the optimal clipping bound $C$ and batch size $B$ in differentially private deep transfer learning.

For clipping bound, various factors such as tighter privacy budgets, less capable pretrained backbones and limited compute—which can be informally characterized as learning problem difficulty—cause a shift in the distribution of gradient norms: gradients grow larger and more variable, especially for harder examples. In these cases, larger clipping bounds, despite injecting more noise per step, better preserve the optimization signal and result in higher overall accuracy. Our experiments across privacy levels, model sizes, datasets, and training budgets confirm this: hard examples become increasingly dominant under high difficulty, and training benefits from a higher clipping threshold to avoid disproportionately discarding their gradients.

In batch size, we have focused on fixed-epoch training where compute constraints limit the total number of samples processed. We have demonstrated that existing guidelines for tuning the batch size that focus on fixed-steps setting where the required compute grows with batch size fail to provide useful guidance in this setting. To fill this gap, we have proposed a combination of minimum number of steps and minimizing the batch size under (nearly) optimal cumulative noise as an effective predictor of optimal batch size across a wide range of settings.

Together, these findings challenge the common practice of fixing $C$ and $B$ across tasks. Fixed hyperparameters ignore how privacy, data, and

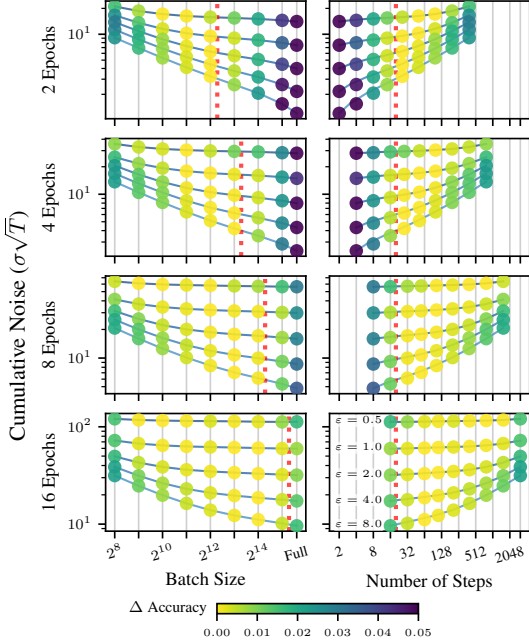

Figure 7: Cumulative noise, batch size and the number of steps under fixed-epoch DP-Adam on CIFAR-100 (ViT-Tiny, FiLM; $N$=50000, $\delta$=$10^{-5}$). Results are averaged over 3 seeds. The learning rate and clipping bound are tuned separately for each point. Left: cumulative noise $\sigma\sqrt{T}$ vs. batch size. Right: the same data shown against the number of steps. The color indicates the difference to best accuracy at each privacy budget ($\varepsilon$) and epoch count; brighter is better. When privacy is tight, cumulative noise exhibits a *noise plateau* where moderate-sized batches can outperform large ones. The vertical lines illustrate that a certain number of steps is required to obtain the optimal accuracy, regardless the number of epochs.

model interact to shape gradient distributions and class separability. By tuning $C$ and $B$ to match learning-problem difficulty, models can learn more effectively and equitably across examples—especially in the high-difficulty regimes where DP training typically struggles. We also found that tuning learning rate, $C$, and $B$ jointly, rather than in isolation, is crucial: for instance, the best learning rate under DP-Adam often scaled with $\sqrt{B}$, echoing non-private scaling rules (You et al., 2020; Malladi et al., 2022).

While our main experiments focus on parameter-efficient (FiLM) fine-tuning for image classification, the underlying mechanisms are general, which we also demonstrated for many settings, such DP-LoRA for images *and* text classification. We also found that when fine-tuning all model parameters on larger models, the average gradient norm grows due to the increased parameter count, preventing the use of smaller clipping bounds. In these settings, both the easy and hard tasks prefer abnormally large clipping bounds.

What our results make clear is that good DP training requires matching the optimizer to the problem: $C$ and $B$ are not just noise parameters, but levers for steering learning. Defaults often suppress the very gradients that matter most, or waste compute on high-variance updates when more steps could be taken at no extra privacy cost. Tuning these hyperparameters can turn good models into great ones, especially in challenging tasks. As DP moves from research into practical deployments, understanding and exploiting these dynamics will be essential for building useful, reliable, and fair private models.

ACKNOWLEDGMENTS

This work was supported by the Research Council of Finland (Flagship programme: Finnish Center for Artificial Intelligence, FCAI, Grant 356499 and Grant 359111), the Strategic Research Council at the Research Council of Finland (Grant 358247) as well as the European Union (Project 101070617). Views and opinions expressed are however those of the author(s) only and do not necessarily reflect those of the European Union or the European Commission. Neither the European Union nor the granting authority can be held responsible for them. The authors wish to thank the CSC – IT Center for Science, Finland for supporting this project with computational and data storage resources. We acknowledge CSC (Finland) for awarding this project access to the LUMI supercomputer, owned by the EuroHPC Joint Undertaking, hosted by CSC (Finland) and the LUMI consortium. The authors acknowledge the research environment provided by ELLIS Institute Finland.

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

## A    LLM USAGE

We used large language models (LLMs), including OpenAI's ChatGPT and GitHub Copilot, at various points during the development of this paper.

These tools assisted with grammar and phrasing, clarification of technical concepts, code generation and refactoring code for data processing, filtering, and visualization, as well as interpretation of intermediate results, such as analyzing plots and drafting captions.

LLMs were especially helpful for rapidly prototyping visualizations by providing raw data and iterating on plotting ideas interactively at the drafting phase. In several cases, they were used to generate the start version of the code, and then reviewed and modified to ensure correctness. Moreover, all such scripts were reviewed and verified by us. We treated generated code as drafts and retained only what was interpretable, correct, and replicable.

We also used LLMs to assist in writing portions of the experiment code, such as data collection logic in our experiment-running system. The core logic was written by the authors and all code was reviewed, tested, and finalized by us.

We also experimented with LLM-assisted initial drafting of the related work section, but due to factual inaccuracies and limited coverage, we abandoned the LLM-generated content and wrote the section manually. We did not use any LLM to design experiments, develop hypotheses, or formulate scientific claims.

We reviewed and finalized all the contents, and take full responsibility for the submission.

## B    EQUIVALENCE OF NORMALIZED VS. STANDARD DP UPDATES

We analyze with the *standard* (unnormalized) variant when clarity of the DP noise's dependence on $C$ is needed, and we train with the *normalized* variant for simpler hyperparameter tuning. The two are mathematically equivalent reparameterizations.

**Standard (unnormalized) update**    In standard DP-SGD (Abadi et al., 2016), the learning rate $\eta$ and the clipping bound $C$ interact

$$\bar{g}_i = g_i \cdot \min\left(1, \frac{C}{\|g_i\|_2}\right), \quad \theta_{t+1} = \theta_t - \frac{\eta_{\text{std}}}{|B|}\left(\sum_{i \in B} \bar{g}_i + \sigma C \xi\right), \quad \xi \sim \mathcal{N}(0, \mathbf{I}_d),$$

where $\xi \in \mathbb{R}^d$ with $d$ the number of trainable parameters, $\mathbf{I}_d \in \mathbb{R}^{d \times d}$ is the identity, and $|B|$ is the expected batch size.

**Normalized update**    In normalized DP-SGD De et al. (2022), normalization is embedded into clipping

$$\bar{g}_i = g_i \cdot \min\left(\frac{1}{C}, \frac{1}{\|g_i\|_2}\right), \quad \theta_{t+1} = \theta_t - \frac{\eta_{\text{norm}}}{|B|}\left(\sum_{i \in B} \bar{g}_i + \sigma \xi\right).$$

Here, $C$ sets sensitivity, while $\eta_{norm}$ sets the step size.

**Equivalence and accounting**    The two forms are equivalent under the reparameterization $\eta_{\text{norm}} = C\eta_{\text{std}}$. This mapping yields identical updates for the same $(q, T, \sigma, C)$, and the privacy accounting is unchanged. We therefore use the standard form in derivations where explicit $C$-dependence of the DP noise is useful (e.g., Section 5), and the normalized form in experiments for simpler tuning.

## C    EXPERIMENT DETAILS

### C.1    HYPERPARAMETER GRID DETAILS

We fine-tune all image models using FiLM (Perez et al., 2018), following the DP-FiLM setup of Tobaben et al. (2023), where we train only the scale and bias of normalization layers and the

Table A1: Hyperparameter grids used for each dataset and model. Method denotes the training method, e.g. FiLM, or from scratch training. Batch sizes are powers of two, with full-batch included. Learning rates are drawn from geometric ranges (number of values shown in parentheses). Clipping bounds are explicitly enumerated.

| Dataset | Model | Method | Subset Size | Batch Sizes $B$ | Learning Rates $\eta$ (geom.) | Clipping Bounds $C$ |
|---|---|---|---|---|---|---|
| CIFAR-100 | ViT-Tiny | DP-Adam, FiLM | 1.0 | $2^8$–$2^{14}$ + full | [0.001, 0.02] (8) | 1e-5, 1e-4, 1e-3, 1e-2, 0.1, 1, 10 |
| CIFAR-100 | ViT-Base | DP-Adam, FiLM | 1.0 | $2^8$–$2^{14}$ + full | [0.001, 0.02] (8) | 1e-5, 1e-4, 1e-3, 1e-2, 0.1, 1, 10 |
| CIFAR-100 | ResNet-50 | DP-Adam, FiLM | 1.0 | $2^8$–$2^{14}$ + full | [0.0005, 0.01] (10) | 1e-5, 1e-4, 1e-3, 1e-2, 0.1, 1, 10 |
| CIFAR-100 (10%) | ViT-Tiny | DP-Adam, FiLM | 0.1 | $2^8$–$2^{14}$ + full | [0.001, 0.02] (8) | 1e-4, 1e-2, 0.1, 1, 5, 10 |
| CIFAR-100 (10%) | ViT-Base | DP-Adam, FiLM | 0.1 | $2^8$–$2^{14}$ + full | [0.001, 0.02] (8) | 1e-4, 1e-2, 0.1, 1, 5, 10 |
| CIFAR-100 (10%) | ResNet-50 | DP-Adam, FiLM | 0.1 | $2^8$–$2^{14}$ + full | [0.0005, 0.01] (10) | 1e-3, 1e-2, 0.1, 1, 10, 26, 42 |
| SUN397 | ViT-Tiny | DP-Adam, FiLM | 1.0 | $2^8$–$2^{16}$ + full | [0.001, 0.02] (8) | 0.1, 1, 10, 12, 14, 18, 26, 42 |
| SUN397 | ViT-Base | DP-Adam, FiLM | 1.0 | $2^8$–$2^{16}$ + full | [0.001, 0.02] (8) | 0.1, 1, 10, 26, 42 |
| SUN397 | ViT-Tiny | DP-Adam, LoRA | 1.0 | $2^8$–$2^{16}$ + full | [0.005, 0.025] (8) | 0.1, 1, 10, 12, 14, 18, 26, 42 |
| SUN397 | ViT-Base | DP-Adam, LoRA | 1.0 | $2^8$–$2^{16}$ + full | [0.005, 0.025] (8) | 0.1, 1, 10, 26, 42 |
| SUN397 | ViT-Tiny | DP-Adam, Full fine-tune | 1.0 | $2^8$–$2^{12}$ + full | [0.0001, 0.002] (10) | 1e-3, 1e-2, 0.1, 1, 10, 25, 29.5, 34.22, 40.03, 46.84, 54.79, 64.11, 75.0 |
| SUN397 | ViT-Base | DP-Adam, Full fine-tune | 1.0 | $2^8$–$2^{12}$ + full | [0.0001, 0.002] (10) | 1e-3, 1e-2, 0.1, 1, 10, 25, 29.5, 34.22, 40.03, 46.84, 54.79, 64.11, 75.0 |
| Cassava | ViT-Tiny | DP-Adam, FiLM | 1.0 | $2^8$–$2^{14}$ + full | [0.001, 0.02] (8) | 0.1, 1, 10, 26, 42 |
| Cassava | ViT-Base | DP-Adam, FiLM | 1.0 | $2^8$–$2^{14}$ + full | [0.001, 0.02] (8) | 0.1, 1, 10, 26, 42 |
| 20 Newsgroups | DistilBERT | DP-Adam, LoRA | 1.0 | $2^8$–$2^{13}$ + full | [0.001, 0.02] (8) | 1e-3, 1e-2, 0.1, 1, 10, 42 |
| CIFAR-10 | WRN-16-4 | DP-Adam, From scratch | 1.0 | $2^8$–$2^{12}$ + full | [0.0001, 0.01] (10) | 1e-5, 1, 10, 50 |

classification head. Following standard practice, we fine-tune language models with LoRA (Hu et al., 2021a), using rank 16 adapters on the query and key matrices of all attention layers. We use a maximum sequence length of 512, which is the DistilBERT (Sanh et al., 2019) default.

We perform a structured grid search over three hyperparameters: learning rate, batch size, and clipping bound. For each model and dataset, we define a geometric learning rate range (with number of values in parentheses), a discrete set of clipping bounds, and a power-of-two batch size sweep up to full-batch training. Subset size refers to the fraction of the training set used.

For datasets that include a validation split, we merge the training and validation sets prior to any subsampling, and train on the combined set. Final performance is reported on the test set.

We do not report all evaluated clipping bounds in the main paper; values unimportant for our results are omitted for clarity.

## C.2 DATASETS

We use four standard image classification datasets: **CIFAR-100**, **CIFAR-100 (10%)**, **SUN397**, and **Cassava**. We also evaluate 20 Newsgroups Mitchell (1997) text classification dataset. All are publicly available through either the Hugging Face `datasets` library or `tensorflow_datasets`. We use canonical train/validation/test splits, and when a separate validation set exists, we merge it into the training set.

Table A2: Public datasets used in our experiments. All are available via Hugging Face or TensorFlow Datasets.

| Dataset | Source | Description |
|---|---|---|
| CIFAR-100 | Krizhevsky & Hinton (2009) | Balanced dataset of natural images with 100 classes and 50k training examples. |
| CIFAR-100 (10%) | Krizhevsky & Hinton (2009) | Stratified 10% random subset of CIFAR-100, used to study limited data regimes. |
| SUN397 | Xiao et al. (2010) | Large-scale scene recognition dataset with 397 classes and roughly 87k training images. |
| Cassava | Mwebaze et al. (2019) | Imbalanced plant disease classification dataset with 5 classes and $\sim$6k examples. Considered out-of-distribution relative to ImageNet pretraining. |
| 20 Newsgroups | Mitchell (1997) | Balanced text classification dataset of roughly 18k Usenet posts across 20 topics, commonly used for benchmarking document categorization. |

## C.3 EXPERIMENT ENVIRONMENT

Our experiments are performed on the clusters, which equipped with AMD EPYC Trento CPU and AMD MI250x GPU.

## C.4 RETAINED WEIGHT

For Fig. 4 (top), we train the DP model with the best hyperparameters under three different clipping bounds, and use $C = 1$ as a baseline. Models are evaluated in `eval` mode.

To analyze gradient clipping, we capture the classifier activations and compute per-sample gradients via a backward pass, and calculate retained weight according to Eq. (3). In the plots, classes are ordered by per-class accuracy. All results use seed 1.

## D KOLOSKOVA ET AL. (2023) BOUND

**Koloskova et al. (2023) bound**   The full bound on the expected gradient norm from Koloskova et al. (2023) is.

$$\mathcal{O}\left(\frac{L\eta}{C}\sigma_{DP}^2. + \sqrt{L\eta\sigma_{DP}}. + \min\{\sigma_B^2, \frac{\sigma_B^4}{C^2}\}. + \eta L\frac{\sigma_B^2}{B}. + \frac{F_0}{\eta T}. + \frac{F_0^2}{\eta^2 T^2 C^2}\right), \qquad (4)$$

where $F_0 = \mathcal{L}(\theta_0) - \mathcal{L}(\theta^*)$ is the difference in loss evaluated at initial weights and at the optimal weights. Denoting $\sigma_{DP} = C\sigma$ makes the dependence on clipping bound (and noise level) explicit.

As the bound depends on the unknown optimal weights, any derived bound with respect to $C$ cannot be determined in advance. Moreover, in contrast to our work, this bound does not model how the gradient norm distributions change as the privacy level changes.

## E OMITTED PROOFS

**Optimal clipping from MSE**   For convenience, we first restate the theorem and then give the proof.

As far as we know, optimal clipping in the sense of minimizing the MSE has not been derived in any existing work. The closest existing work we are aware of is Amin et al. (2019), who derive an upper bound for MSE for pure DP ($\delta = 0$) using Laplace noise. We instead focus on the Gaussian mechanism, and derive exact MSE, not an upper bound.

**Assumption 5.1.** *Assume there is no mini-batch sampling noise, we use standard per-sample constant clipping with $C \in [\min_i \|g_i\|, \max_i \|g_i\|]$, and the Gaussian mechanism with noise level $\sigma$ to privatize the sum of clipped gradients.*

**Theorem 5.2.** *Under Assumption 5.1, an optimal clipping constant $C^*$ that minimizes the mean squared error between the per-sample clipped DP gradient $\tilde{g}$ and the true gradient $g$ for a fixed mini-batch satisfies*

$$C^* = \begin{cases} \|g_i\| \text{ for some } i, \text{ or} \\ \frac{N_{C^*}^T G_{C^*}}{N_{C^*}^T N_{C^*} + \sigma^2 d}, \end{cases} \tag{1}$$

*where $d$ is the dimensionality of the gradient, $G_C := \sum_{i \in I_C} g_i$ and $N_C := \sum_{i \in I_C} \frac{g_i}{\|g_i\|}$, and $I_C = \{i : \|g_i\| > C\}$ denotes the indices of the clipped gradients.*

*Proof.* Let $g_i, i = 1, \ldots, B$ denote the true per-example gradients in a given mini-batch, $\bar{g}_i$ the corresponding clipped gradients, and $\xi$ the DP noise. The mean squared error of the total clipped noisy mini-batch gradient as a function of $C$ is

$$\text{MSE}(C) := \mathbb{E} \left\| \sum_i \bar{g}_i + \xi - \sum_i g_i \right\|^2 = \mathbb{E}\|\xi\|^2 + \left\| \sum_i \bar{g}_i - \sum_i g_i \right\|^2$$

$$= C^2 \sigma^2 d + \left\| \sum_{i:\|g_i\|>C} \frac{\|g_i\| - C}{\|g_i\|} g_i \right\|^2 = C^2 \sigma^2 d + \left\| \sum_{i \in I_C} \frac{\|g_i\| - C}{\|g_i\|} g_i \right\|^2, \tag{5}$$

where $I_C = \{i : \|g_i\| > C\}$ denotes the indices of the clipped gradients. Noting that $\text{MSE}(C)$ is a continuous function defined on a bounded interval $[\min_i \|g_i\|, \max_i \|g_i\|]$, there exists a minimizer $C^*$. If $C^* \neq \|g_i\|$ for all $i$, then $C^* \in (\|g_i\|, \|g_j\|)$ for some $i, j$. On this open interval, MSE is differentiable with the derivative

$$\frac{d\text{MSE}(C)}{dC} = 2C\sigma^2 d + 2 \left( -\sum_{i \in I_C} \frac{g_i}{\|g_i\|} \right)^T \left( \sum_{i \in I_C} \frac{\|g_i\| - C}{\|g_i\|} g_i \right)$$

$$= 2C\sigma^2 d - 2 \left( \sum_{i \in I_C} \frac{g_i}{\|g_i\|} \right)^T \left( \sum_{i \in I_C} g_i \right) + 2C \left( \sum_{i \in I_C} \frac{g_i}{\|g_i\|} \right)^T \left( \sum_{i \in I_C} \frac{g_i}{\|g_i\|} \right) \tag{6}$$

By denoting $G_C := \sum_{i \in I_C} g_i$ and $N_C := \sum_{i \in I_C} \frac{g_i}{\|g_i\|}$, this simplifies to

$$\frac{d\text{MSE}(C)}{dC} = 2C\sigma^2 d + 2C N_C^T N_C - 2N_C^T G_C \tag{7}$$

The zero of the derivative $C^*$ satisfies

$$C^* = \frac{N_{C^*}^T G_{C^*}}{N_{C^*}^T N_{C^*} + \sigma^2 d} t, \tag{8}$$

which concludes the proof. $\qquad \square$

**Connecting MSE to optimization progress** We start by stating some helpful standard results, then restate the theorem and finally continue with the proof.

**Definition E.1** (Smoothness). *A differentiable function $f$ is L-smooth, if*

$$\|\nabla f(x) - \nabla f(y)\| \leq L\|x - y\|, \ \forall x, y \in \mathbb{R}^d. \tag{9}$$

Assuming smoothness, we have the well-known quadratic upper bound given next in Lemma E.1.

**Lemma E.1** (Quadratic upper bound). *Assume $f : \Omega \to \mathbb{R}$ is L-smooth on a convex domain $\Omega$. Then*

$$f(y) \leq f(x) + \langle \nabla f(x), y - x \rangle + \frac{L}{2}\|y - x\|_2^2. \tag{10}$$

*Proof.* See any standard lecture notes on smooth optimization. $\qquad \square$

**Assumption 5.3.** *Assume an L-smooth loss function $\mathcal{L}$ (Definition E.1), and that we want to minimize $\mathcal{L}$ via gradient descent, using step size $\eta \leq \frac{1}{L}$, and the per-sample clipped and noisy DP gradients $\tilde{g}_t = \sum_i \bar{g}_i^{(t)} + \xi^{(t)}$ at step $t = 1, \ldots, T$ instead of the true gradients $g_t = \sum_i g_i^{(t)} = \nabla\mathcal{L}(\theta_t)$.*

**Theorem 5.4.** *Under Assumption 5.3, the expected improvement in the loss from step $t$ to $t + 1$ is bounded as*

$$\mathbb{E}[\mathcal{L}(\theta_{t+1})|\theta_t] \leq \mathcal{L}(\theta_t) - \frac{\eta}{2}\|\nabla\mathcal{L}(\theta_t)\|_2^2 + \frac{\eta}{2}\text{MSE}_t(C) \tag{2}$$

*Proof.* Let $e_t(C) = \tilde{g}_t - g_t$, i.e., the error when using $C$ for the per-sample clipping at step $t$, so $\mathbb{E}[\|e_t(C)\|_2^2]$ is $\text{MSE}_t(C)$ as defined in Eq. (5) at step $t$. In the following, we write $e_t := e_t(C)$ for brevity without any risk of confusion.

We have

$$\mathcal{L}(\theta_{t+1}) \leq \mathcal{L}(\theta_t) + \langle g_t, \theta_{t+1} - \theta_t \rangle + \frac{L}{2}\|\theta_{t+1} - \theta_t\|_2^2 \tag{11}$$

$$\leq \mathcal{L}(\theta_t) - \eta\langle g_t, g_t + e_t \rangle + \frac{L\eta^2}{2}\|g_t + e_t\|_2^2 \tag{12}$$

$$\leq \mathcal{L}(\theta_t) - \eta\|g_t\|_2^2 - \eta\langle g_t, e_t \rangle + \frac{L\eta^2}{2}\left(\|g_t\|_2^2 + 2\langle g_t, e_t \rangle + \|e_t\|_2^2\right) \tag{13}$$

$$= \mathcal{L}(\theta_t) + \left(\frac{L\eta^2}{2} - \eta\right)\|g_t\|_2^2 + \left(L\eta^2 - \eta\right)\langle g_t, e_t \rangle + \frac{L\eta^2}{2}\|e_t\|_2^2 \tag{14}$$

$$\leq \mathcal{L}(\theta_t) + \left(\frac{L\eta^2}{2} - \eta\right)\|g_t\|_2^2 + |\left(L\eta^2 - \eta\right)||\langle g_t, e_t \rangle| + \frac{L\eta^2}{2}\|e_t\|_2^2, \tag{15}$$

where we first use Lemma E.1 with the assumption that $g_t = \nabla\mathcal{L}(\theta_t)$, then note that $\theta_{t+1} - \theta_t = -\eta(g_t + e_t)$, and expand the terms to get Eq. (13).

To bound the cross-terms, we have $|\langle g_t, e_t \rangle| \leq \|g_t\|_2\|e_t\|_2 \leq \frac{1}{2}\|g_t\|_2^2 + \frac{1}{2}\|e_t\|_2^2$, due to Cauchy-Schwarz and the elementary inequality $2ab \leq a^2 + b^2$. Since we assume $\eta \leq 1/L \Leftrightarrow \eta L \leq 1$, we have $L\eta^2 - \eta = \eta(L\eta - 1) \leq 0$, and therefore $|L\eta^2 - \eta| = \eta - L\eta^2$. Continuing from Eq. (15),

$$\mathcal{L}(\theta_{t+1}) \leq \mathcal{L}(\theta_t) + \left(\frac{L\eta^2}{2} - \eta\right)\|g_t\|_2^2 + \left(\eta - L\eta^2\right)\frac{\|g_t\|_2^2 + \|e_t\|_2^2}{2} + \frac{L\eta^2}{2}\|e_t\|_2^2 \tag{16}$$

$$= \mathcal{L}(\theta_t) - \frac{\eta}{2}\|g_t\|_2^2 + \frac{\eta}{2}\|e_t\|_2^2. \tag{17}$$

Taking expectations on both sides, writing the current parameter value $\theta_t$ explicitly as conditioning the expectation, and using the assumption that $g_t = \nabla\mathcal{L}(\theta_t)$ we finally have

$$\mathbb{E}[\mathcal{L}(\theta_{t+1})|\theta_t] \leq \mathcal{L}(\theta_t) - \frac{\eta}{2}\|\nabla\mathcal{L}(\theta_t)\|_2^2 + \frac{\eta}{2}\text{MSE}_t(C), \tag{18}$$

which concludes the proof. $\square$

# F ACCURACY BY MODEL AND BY PRIVACY BUDGET

## F.1 BY MODEL (FIXED PRIVACY BUDGET)

**Dataset: CIFAR100 (10%)**

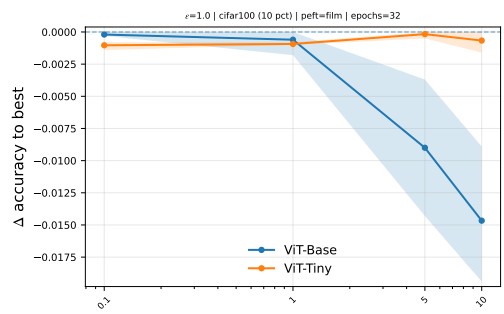 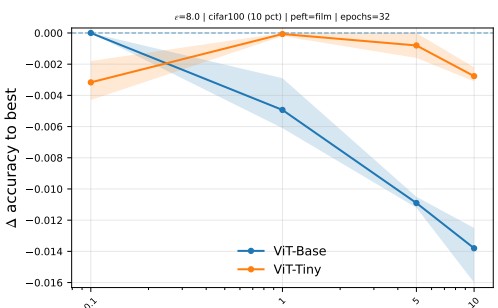

(a) Clipping bound vs. accuracy comparing ViT-Base and ViT-Tiny on CIFAR-100 (10% subset, 32 epochs, $\delta = 10^{-5}$) at $\epsilon = 1.0$.

(b) Clipping bound vs. accuracy comparing ViT-Base and ViT-Tiny on CIFAR-100 (10% subset, 32 epochs, $\delta = 10^{-5}$) at $\epsilon = 8.0$.

Figure A1: Clipping bound vs. accuracy comparison on CIFAR-100 (10% subset)

**Dataset: CASSAVA**

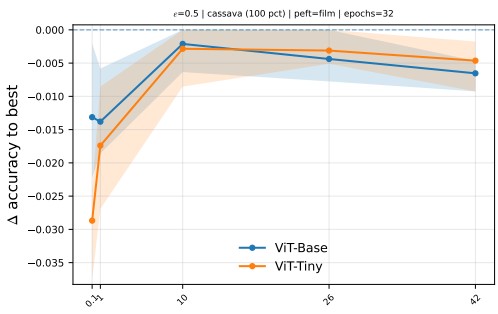 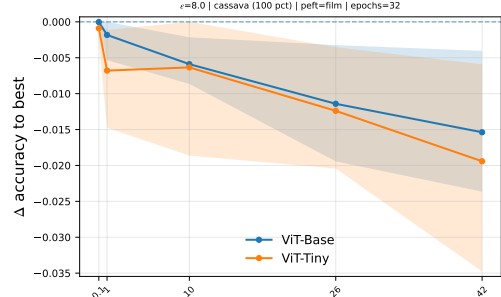

(a) Clipping bound vs. accuracy comparing ViT-Base and ViT-Tiny on Cassava (full dataset, 32 epochs, $\delta = 10^{-5}$) at $\epsilon = 0.5$.

(b) Clipping bound vs. accuracy comparing ViT-Base and ViT-Tiny on Cassava (full dataset, 32 epochs, $\delta = 10^{-5}$) at $\epsilon = 8.0$.

Figure A2: Clipping bound vs. accuracy comparison on Cassava

**Dataset: CIFAR100 (100%)**

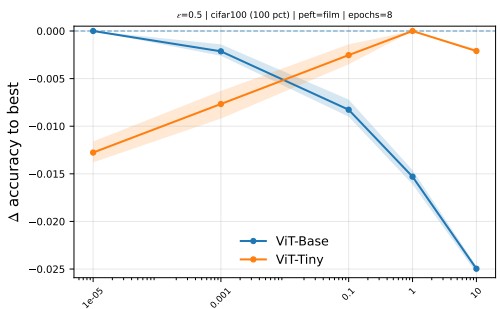 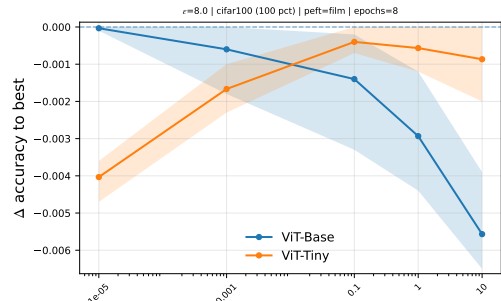

(a) Clipping bound vs. accuracy comparing ViT-Base and ViT-Tiny on CIFAR-100 (full dataset, 8 epochs, $\delta = 10^{-5}$) at $\epsilon = 0.5$.

(b) Clipping bound vs. accuracy comparing ViT-Base and ViT-Tiny on CIFAR-100 (full dataset, 8 epochs, $\delta = 10^{-5}$) at $\epsilon = 8.0$.

Figure A3: Clipping bound vs. accuracy comparison on CIFAR100 (100%)

**Dataset: SUN397**

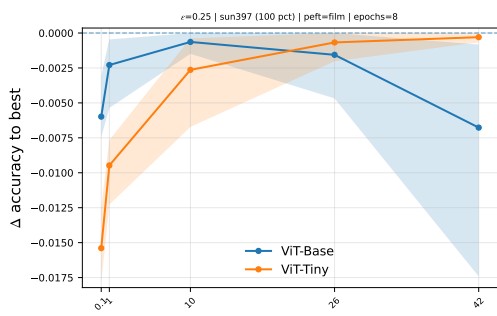 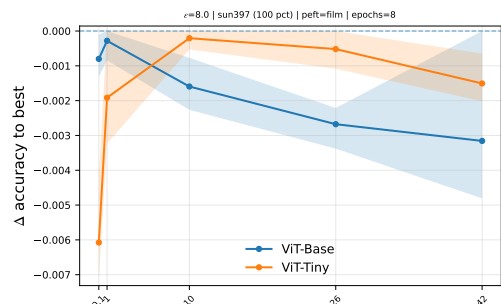

(a) Clipping bound vs. accuracy comparing ViT-Base and ViT-Tiny on SUN397 (full dataset, 8 epochs, $\delta = 10^{-5}$) at $\epsilon = 0.25$.

(b) Clipping bound vs. accuracy comparing ViT-Base and ViT-Tiny on SUN397 (full dataset, 8 epochs, $\delta = 10^{-5}$) at $\epsilon = 8.0$.

Figure A4: Clipping bound vs. accuracy comparison on SUN397

## F.2 BY PRIVACY BUDGET (FIXED MODEL)

### Dataset: CIFAR100 (10%)

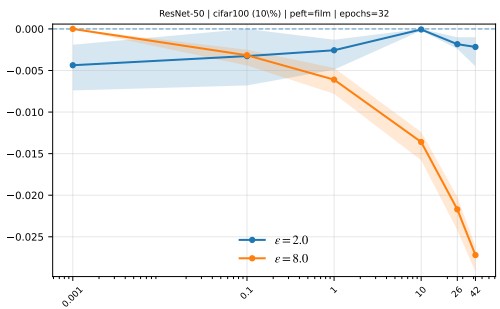
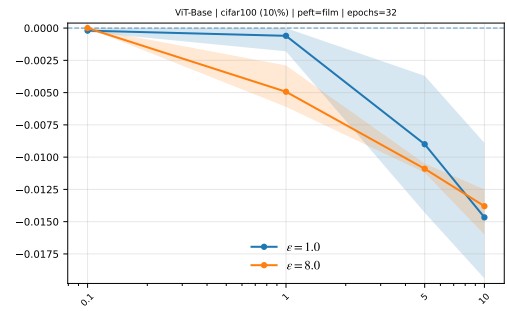

(a) Clipping bound vs. accuracy for ResNet-50 on CIFAR-100 (10% subset, 32 epochs, $\delta = 10^{-5}$). Privacy settings: $\epsilon \in \{2, 8\}$.

(b) Clipping bound vs. accuracy for ViT-Base on CIFAR-100 (10% subset, 32 epochs, $\delta = 10^{-5}$). Privacy settings: $\epsilon \in \{1, 8\}$.

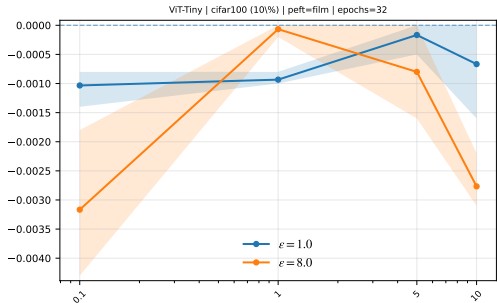

(a) Clipping bound vs. accuracy for ViT-Tiny on CIFAR-100 (10% subset, 32 epochs, $\delta = 10^{-5}$). Privacy settings: $\epsilon \in \{1, 8\}$.

Figure A6: Clipping bound vs. accuracy comparison on CIFAR100 (10%).

**Dataset: CASSAVA**

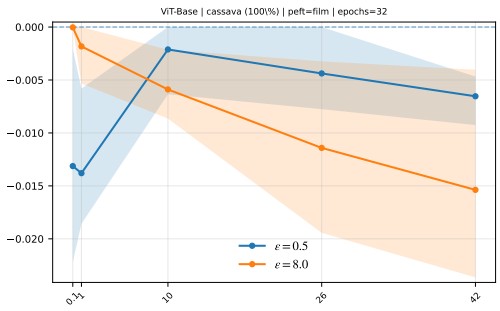
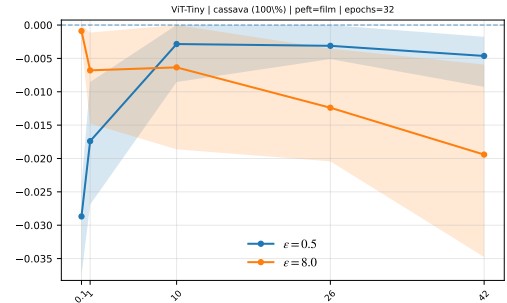

(a) Clipping bound vs. accuracy for ViT-Base on Cassava (full dataset, 32 epochs, $\delta = 10^{-5}$). Privacy settings: $\epsilon \in \{0.25, 8\}$.

(b) Clipping bound vs. accuracy for ViT-Tiny on Cassava (full dataset, 32 epochs, $\delta = 10^{-5}$). Privacy settings: $\epsilon \in \{0.25, 8\}$.

Figure A7: Clipping bound vs. accuracy comparison on CASSAVA.

**Dataset: CIFAR100**

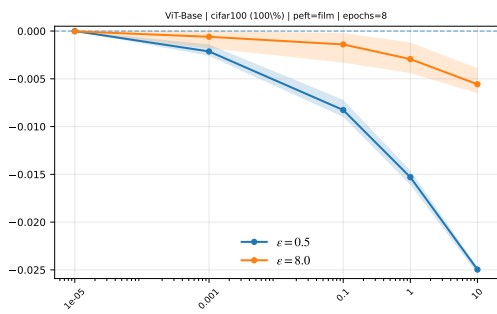
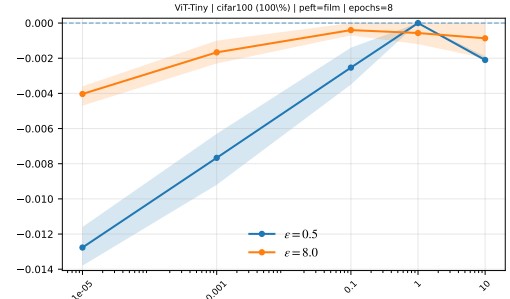

(a) Clipping bound vs. accuracy for ViT-Base on CIFAR-100 (full dataset, 8 epochs, $\delta = 10^{-5}$). Privacy settings: $\epsilon \in \{0.5, 8\}$.

(b) Clipping bound vs. accuracy for ViT-Tiny on CIFAR-100 (full dataset, 8 epochs, $\delta = 10^{-5}$). Privacy settings: $\epsilon \in \{0.5, 8\}$.

Figure A8: Clipping bound vs. accuracy comparison on CIFAR100 (100%).

**Dataset: SUN397**

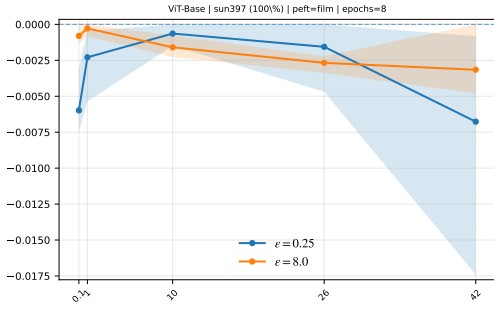
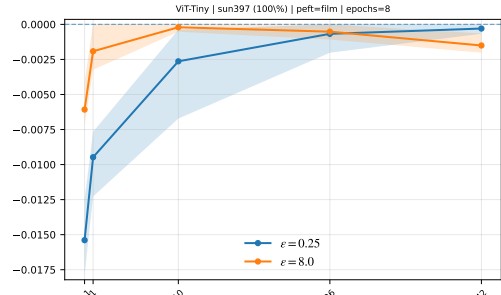

(a) Clipping bound vs. accuracy for ViT-Base on SUN397 (full dataset, 8 epochs, $\delta = 10^{-5}$). Privacy settings: $\epsilon \in \{0.25, 8\}$.

(b) Clipping bound vs. accuracy for ViT-Tiny on SUN397 (full dataset, 8 epochs, $\delta = 10^{-5}$). Privacy settings: $\epsilon \in \{0.25, 8\}$.

Figure A9: Clipping bound vs. accuracy comparison on SUN397.

# G FINE-TUNING WITH LoRA

In this appendix, we will study fine-tuning with LoRA and its effects on the clipping bound.

## G.1 IMAGE CLASSIFICATION WITH LoRA

We evaluate fine-tuning image models with LoRA. We fine-tune ViT-Tiny ($\varepsilon$=1, hard) and ViT-Base ($\varepsilon$=8, easy) on SUN397 using rank-1 LoRA adapters on all attention layers, training only the LoRA parameters and the classifier head. After performing an initial sweep over LoRA ranks $1, 2, 4, 8, 16$ while running HPO on the other parameters, we opter for rank-1 LoRA adapters as they gave the best performance (see Table A3.

We use DP-Adam with the normalized update, Poisson subsampling, PRV accounting, and $\delta = 10^{-5}$, running for 8 epochs. We jointly tune learning rate, batch size, and clipping bound for each case (see Appendix C.1 for details).

Fig. A10 shows the effect of the clipping bound on test accuracy. In the easy setting (ViT-Base, $\varepsilon = 8$), performance is remarkably stable across a wide range of clipping values, with small $C$ performing best. In contrast, in the hard setting (ViT-Tiny, $\varepsilon = 1$) benefits from larger clipping bounds, with accuracy improving as $C$ increases. Under higher learning-problem difficulty, the optimal clipping bound shifts upward.

To further analyze if the findings match our analysis, in Fig. A11 we plot the per-class relative retained weight and the resulting per-class accuracy change. In the easy ViT-Base case, all clipping bounds result in remarkably stable accuracy. However, in the hard (ViT-Tiny, $\varepsilon$=1 case, small clipping bounds reduce retained weight for the hardest classes, while $C = 10$ preserves more gradient signal and improves their accuracy.

Finally, Fig. A12 provides the full accuracy grid over $(B, C)$ for both models.

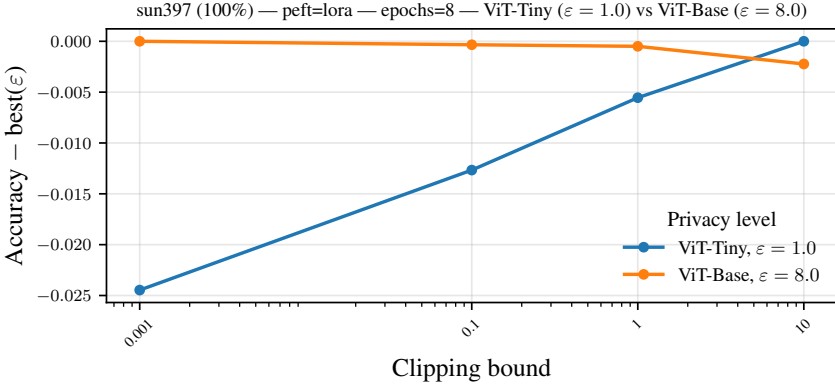

Figure A10: Effect of clipping bound on image classification with LoRA. Accuracy difference to the best clipping bound at each $\varepsilon$ for SUN397 fine-tuning with rank 1 Lora adapters, 8 epochs, $\delta = 10^{-5}$. The hard configuration (ViT-Tiny, $\varepsilon = 1$) benefits from larger $C$, while the easy configuration (ViT-Base, $\varepsilon = 8$) clearly prefers a smaller clipping bound.

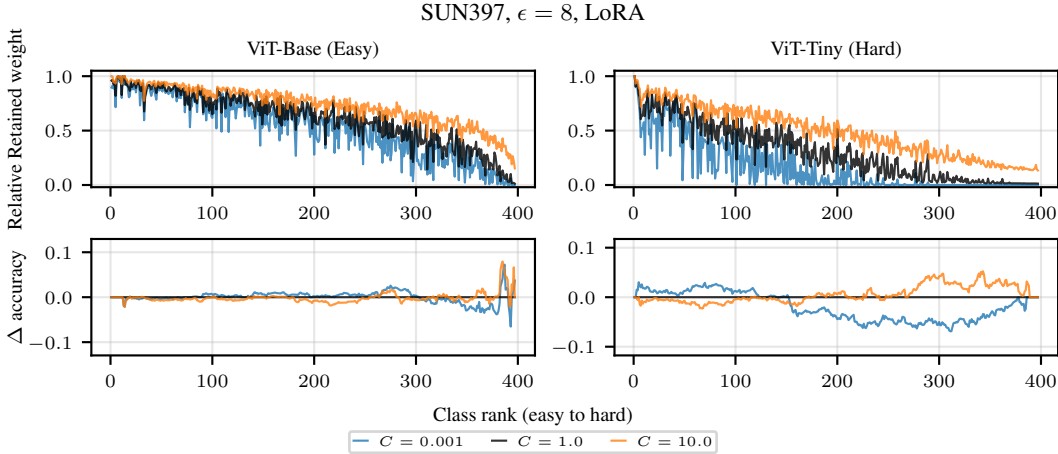

Figure A11: Per-class relative retained weight and per-class accuracy change for SUN397 with LoRA rank 1 adapters, 8 epochs, $\delta = 10^{-5}$, and $\varepsilon = 8$. Left: ViT-Base, $\varepsilon$=8(easy); right: ViT-Tiny, $\varepsilon$=1 (hard). Classes are ordered from easy to hard by per-class accuracy. Top: relative retained weight. Bottom: per-class accuracy difference relative to $C = 1$. For ViT-Tiny, small $C$ reduces retained weight for hard classes and degrades their accuracy, while $C = 10$ preserves their gradients and improves performance.

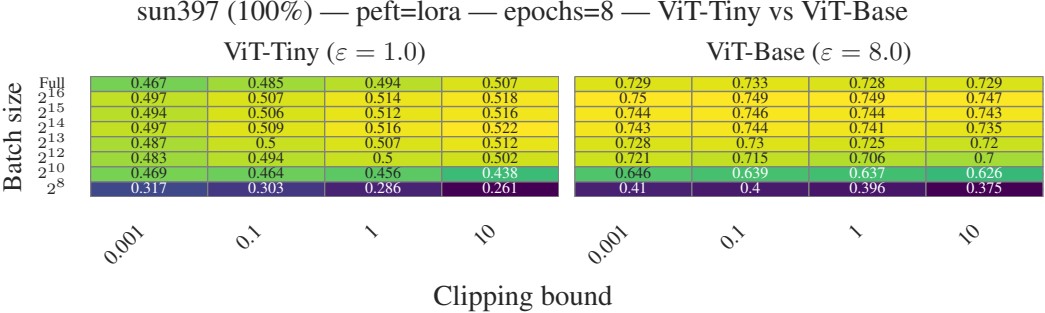

Figure A12: Accuracy for each configuration of batch size and clipping bound when fine-tuning SUN397 with DP LoRA (8 epochs, $\delta = 10^{-5}$). Left: ViT-Tiny with $\varepsilon = 1$; right: ViT-Base with $\varepsilon = 8$. In the hard regime, the best configurations consistently use larger clipping bounds ($C \approx 10$), whereas the easy regime is remarkable stable across $C$'s (see Fig. A10).

## G.2    TEXT CLASSIFICATION WITH LORA

We further validate our observations on a text classification task using parameter-efficient fine-tuning. Following standard practices, we fine-tune DistilBERT Sanh et al. (2019) on the 20 Newsgroups dataset Mitchell (1997) using LoRA with rank-16 adapters (see Appendix C.1 for details) for 8 epochs. We use maximum sequence length of $512$ (model default). We use DP-Adam with the normalized update, Poisson subsampling, and $\delta = 10^{-5}$ while varying the clipping bound $C$ and privacy budget $\varepsilon$.

Fig. A13 shows that, although the absolute accuracy differences between clipping bounds are small on this task, different clipping choices still affect performance as our theory suggests. Fig. A14 explains these differences using relative retained weights, revealing the same underlying mechanism: small clipping bound suppresses signal from hard examples, especially under higher learning-problem difficulty. We speculate that the accuracy differences would become larger with a more difficult task or if switching to a lower capability pretrained model.

Finally, Fig. A15 reports the full accuracy grid over all configurations. We always tune the learning rate separately for each configuration.

Table A3: Test accuracy of LoRA with sweep over ranks. ViT-Tiny on SUN397, 8 epochs, $\delta=10^{-5}$. Each row shows the accuracy results from the best configuration found by HPO over learning rate, batch size, with clipping bound fixed to 1 to reduce HPO costs. Higher ranks can degrade performance, especially under tighter privacy.

| LoRA rank | Accuracy at $\varepsilon=0.5$ | Accuracy at $\varepsilon=8$ |
|---|---|---|
| 1 | 0.4107 | 0.6512 |
| 2 | 0.4020 | 0.6514 |
| 4 | 0.4055 | 0.6535 |
| 8 | 0.3924 | 0.6482 |
| 16 | 0.3800 | 0.6481 |

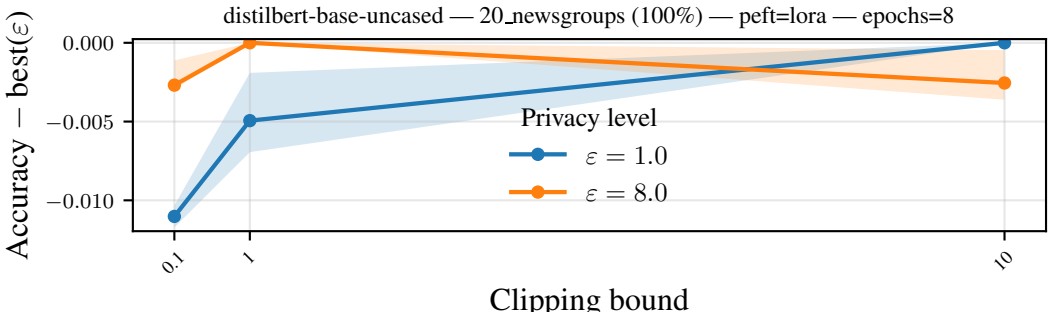

Figure A13: Effect of clipping bound on text classification with LoRA adapters. Accuracy difference to the best clipping bound for each privacy level, DistilBERT Sanh et al. (2019) on 20 Newsgroups dataset Mitchell (1997) with rank 16 LoRA adapters, 8 epochs, and $\delta = 10^{-5}$. Lines indicate median over three repeats with min–max bands. Tighter privacy ($\varepsilon = 1$) shifts the optimal clipping bound towards larger $C$, while looser privacy ($\varepsilon = 8$) prefers a smaller bound, consistent with our findings.

## H  AUTOMATIC CLIPPING COMPARISON

We compare against the automatic clipping method proposed by Bu et al. (2023). Specifically, we use the `AUTO-S` variant, which the authors report to outperform `AUTO-V`, and set $\gamma = 0.01$ as recommended by the authors. The results in Table A4 correspond to those shown in Fig. 5, providing exact accuracy values across datasets and privacy levels.

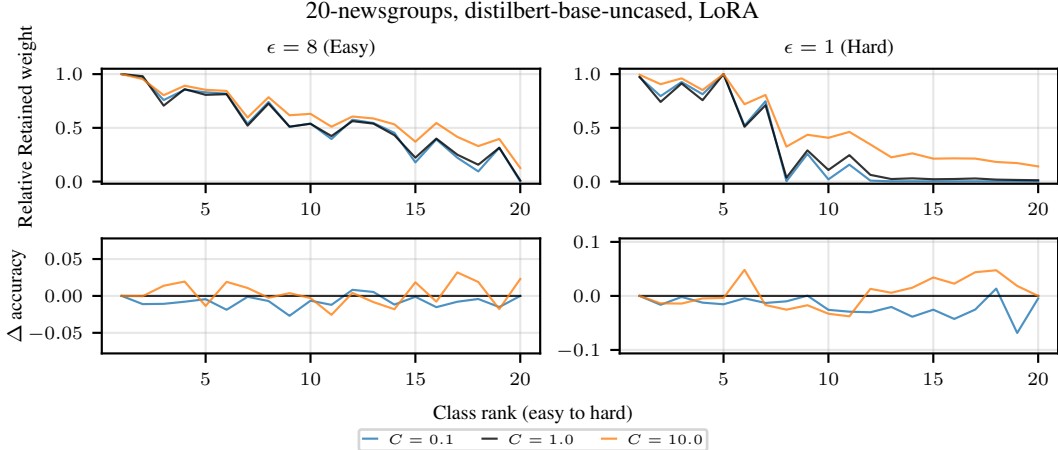

Figure A14: Per-class retained weight and accuracy for text classification with LoRA adapters on 20 Newsgroups dataset Mitchell (1997) using the DistilBERT Sanh et al. (2019), rank 16 LoRA adapters, 8 epochs, $\delta = 10^{-5}$. Classes are ordered from easy to hard by per-class accuracy. The batch size and the learning rate are tuned separately for each case. Top: change in per-class accuracy relative to $C = 1$. Bottom: relative retained weight for different clipping bounds. Under tight privacy ($\varepsilon = 1$), small $C$ over-clips gradients from the hardest classes, while larger $C$ preserves more of their signal and improves their accuracy.

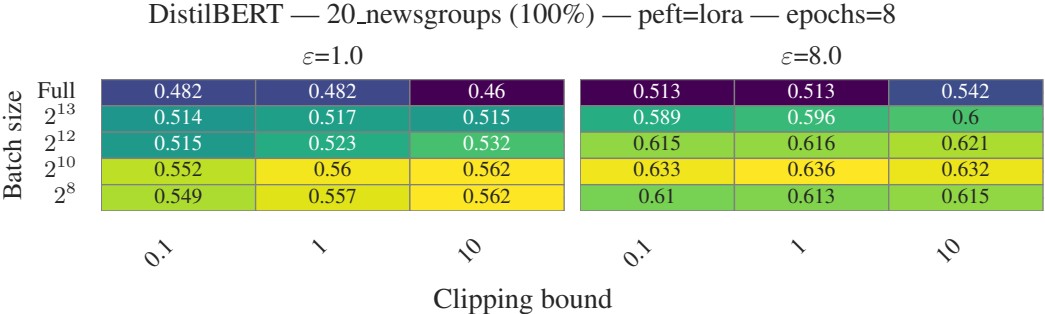

Figure A15: Absolute accuracies achieved with each configuration for DP LoRA fine-tuning on 20 Newsgroups dataset Mitchell (1997) DistilBERT Sanh et al. (2019), rank 16 LoRA adapters, 8 epochs, $\delta = 10^{-5}$. Learning rate is tuned separately for each case.

## I  OTHER TRAINING METHODS

In this appendix, we extend the analysis in Figure 4 by examining the per-class effects of gradient clipping under three additional training regimes: (i) training from scratch, (ii) DP-SGD, and (iii) full-parameter fine-tuning.

For comparability across models and optimizers, we plot the relative retained weight after clipping for each class, normalized by the class with the highest retained weight under the baseline setting ($C=1$). This metric quantifies the fraction of gradient signal preserved after clipping. Flatter and higher curves reflect more uniform gradient preservation across classes, while steep drops indicate that difficult classes lose gradient mass disproportionately. Excessive clipping therefore suppresses learning for these classes, whereas sufficiently large clipping bounds maintain gradient signal even as task difficulty increases.

### I.1  TRAINING FROM SCRATCH WITH DP-ADAM

We examine training from scratch to assess whether the same clipping effects arise in a setting without pre-trained features. We train WideResNet-16-4 (Zagoruyko & Komodakis, 2016) on CIFAR-10

Table A4: Comparison of AUTO-S (with $\gamma = 0.01$) against our best-tuned clipping bound across datasets and privacy levels. Each row shows mean macro accuracy with [min, max] across three random seeds. These are the absolute accuracies for the lines plotted in Fig. 5.

| Dataset | $\varepsilon$ | AUTO-S | Tuned (Ours) |
|---|---|---|---|
| SUN397 | 0.5 | 0.403 [0.399, 0.405] | 0.430 [0.428, 0.434] |
| SUN397 | 1 | 0.510 [0.508, 0.512] | 0.536 [0.534, 0.537] |
| SUN397 | 2 | 0.584 [0.581, 0.586] | 0.601 [0.598, 0.603] |
| SUN397 | 4 | 0.628 [0.624, 0.631] | 0.639 [0.636, 0.642] |
| SUN397 | 8 | 0.650 [0.645, 0.655] | 0.660 [0.655, 0.663] |
| Cassava | 0.5 | 0.604 [0.584, 0.620] | 0.637 [0.629, 0.648] |
| Cassava | 1 | 0.699 [0.691, 0.709] | 0.709 [0.701, 0.718] |
| Cassava | 2 | 0.754 [0.752, 0.756] | 0.759 [0.757, 0.762] |
| Cassava | 4 | 0.795 [0.792, 0.801] | 0.797 [0.792, 0.802] |
| Cassava | 8 | 0.812 [0.799, 0.825] | 0.814 [0.805, 0.827] |
| CIFAR-100 | 0.5 | 0.781 [0.781, 0.782] | 0.786 [0.784, 0.788] |
| CIFAR-100 | 1 | 0.812 [0.811, 0.814] | 0.815 [0.815, 0.817] |
| CIFAR-100 | 2 | 0.830 [0.828, 0.832] | 0.832 [0.831, 0.833] |
| CIFAR-100 | 4 | 0.841 [0.840, 0.841] | 0.842 [0.841, 0.843] |
| CIFAR-100 | 8 | 0.846 [0.845, 0.847] | 0.847 [0.847, 0.848] |

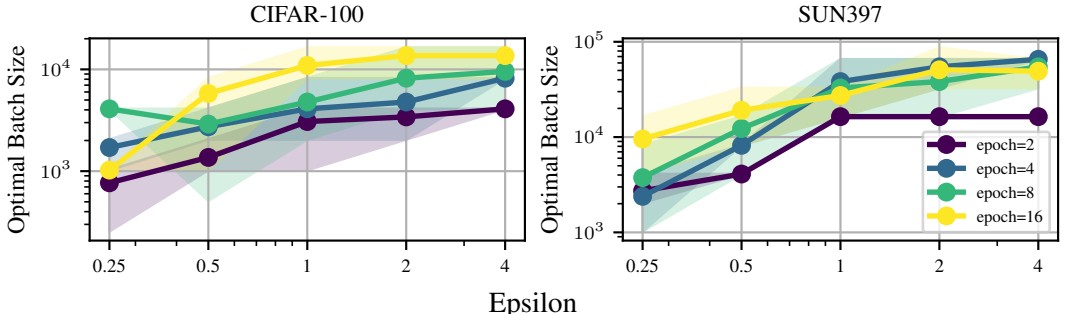

Figure A16: Optimal batch size under fixed-epoch DP-Adam on CIFAR-100 and SUN397 with ViT-Tiny and FiLM ($\delta = 10^{-5}$; $N_{\text{CIFAR-100}} = 50{,}000$, $N_{\text{SUN397}} = 87{,}002$). Curves show means across epochs and $\varepsilon$, with shaded areas indicating variation over 3 seeds. The optimal batch size increases with training epochs, most steeply at low epochs. Smaller $\varepsilon$ favor smaller batches, while gains from very large batches taper off as $\varepsilon$ or epochs grow.

for 8 epochs using DP-Adam without any PEFT method (namely, no parameter-efficient layers; see Appendix C.1 for details), varying the clipping bound $C$ and privacy budget $\varepsilon$. Figure A18 summarizes the accuracies achieved under each hyperparameter configuration.

Despite the simplicity of the task, different clipping bounds still lead to measurable accuracy differences. Figure A17 explains these differences through per-class retained weights. As in our other results, small clipping bounds disproportionately suppress gradients from the hardest classes, while larger bounds preserve more signal and improve their accuracy, especially under higher learning-problem difficulty.

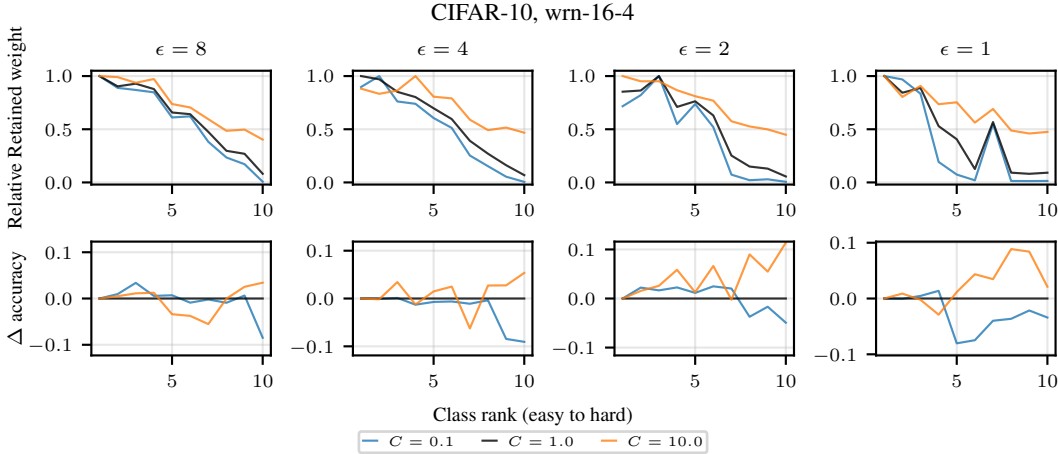

Figure A17: Relative retained gradient weight per class on CIFAR-10, under increasing task difficulty (left to right). The model is WideResNet-16-4, trained for 8 epochs from scratch using DP-Adam and FiLM (Perez et al., 2018); $\delta = 10^{-5}$.

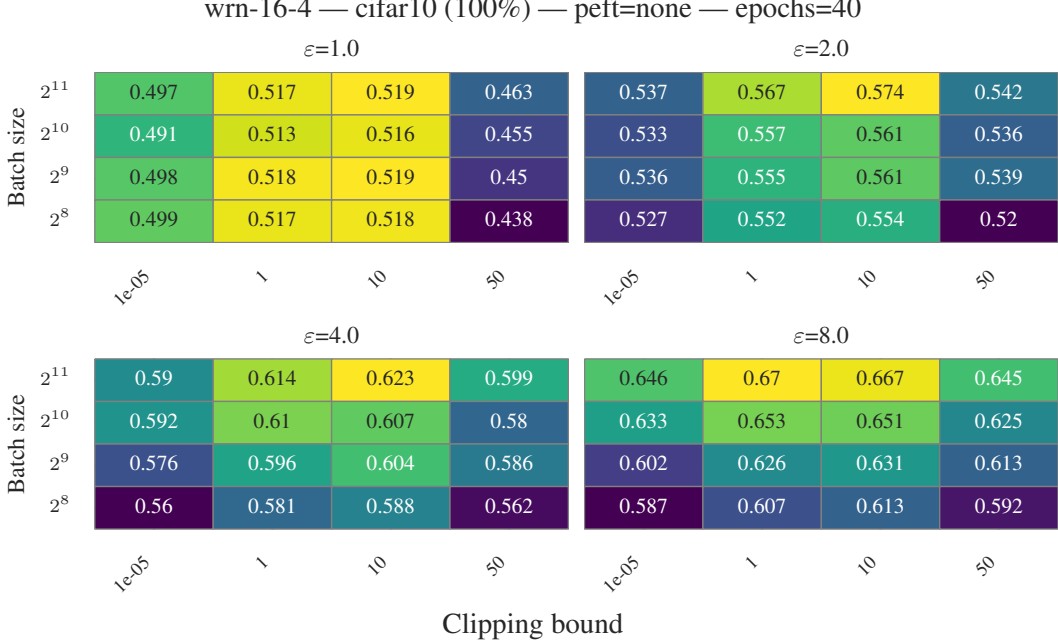

Figure A18: Accuracy achieved with each configuration when training WideResNet-16-4 from scratch on CIFAR-10 with DP-Adam; 8 epochs, $\delta = 10^{-5}$. The batch size and learning rate are tuned separately for each $\varepsilon$ and clipping bound $C$.

## I.2 FINE-TUNING WITH DP-SGD

We evaluate the behavior of clipping under DP-SGD, focusing on ViT-Tiny fine-tuning on SUN397. We train for 8 epochs using DP-SGD with FiLM, Poisson subsampling, and $\delta = 10^{-5}$, while varying the clipping bound $C$ and the privacy budget $\varepsilon$. Figure A20 summarizes the accuracies obtained across the hyperparameter configurations.

Consistent with our earlier findings, Figure A19 shows that small clipping bounds reduce the retained gradient weight for the hardest classes, especially under high learning-problem difficulty. Larger clipping bounds preserve more class-specific gradient signal and thus lead to improved accuracy for these harder classes, demonstrating that the same mechanism appears also with DP-SGD.

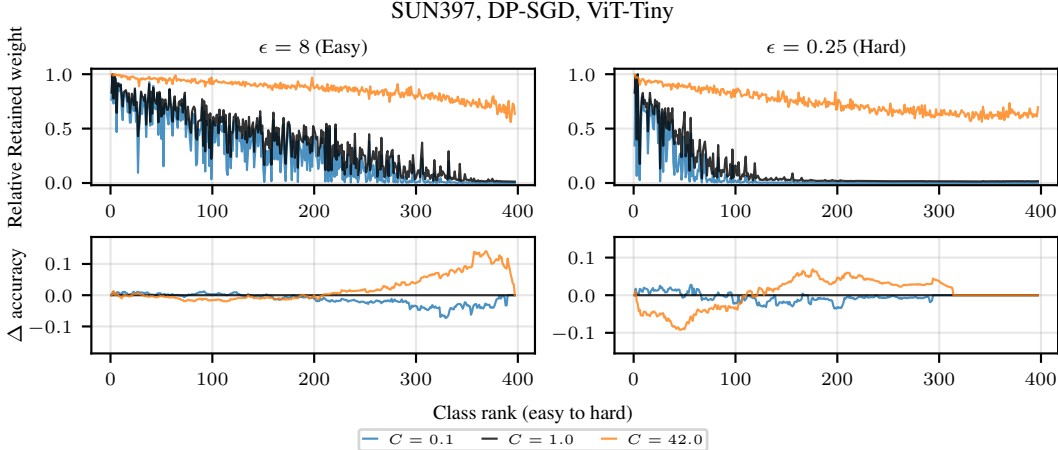

Figure A19: Per-class relative retained gradient weight on SUN397 with ViT-Tiny trained for 8 epochs using DP-SGD with FiLM; $\delta = 10^{-5}$. Classes are ordered from easy to hard by per-class accuracy. Small clipping bounds disproportionately suppress gradients for the hardest classes, while larger bounds maintain more signal and improve their performance under higher task difficulty.

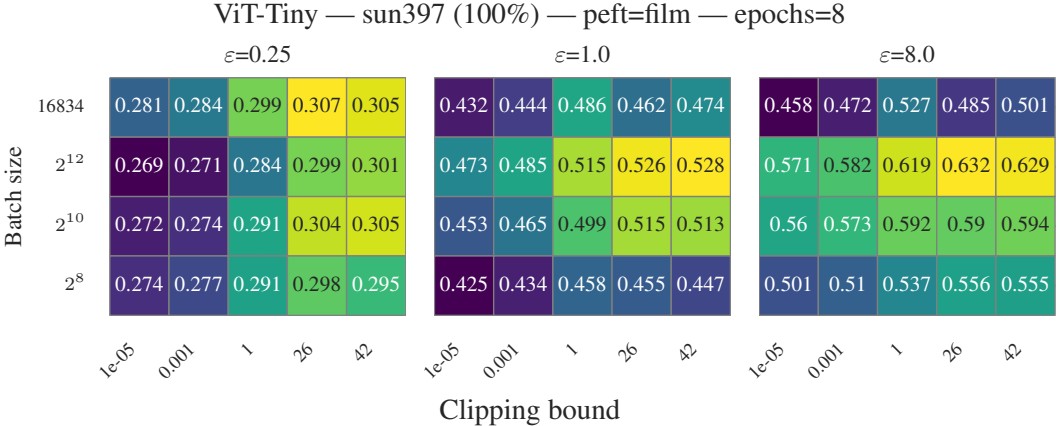

Figure A20: Accuracy across hyperparameter configurations for DP-SGD training. We tune the learning rate separately for each configuration.

## I.3 FULL FINE-TUNE

Here we evaluate if our findings hold for full fine-tune; meaning that instead of employing parameter-efficient fine-tuning (PEFT) method, we fine-tune all the model parameters. We find that full fine-tuning is different from PEFT fine-tuning: the clipping bound does not clearly depend on task difficulty—instead a very large clipping bound seems to be the most effective, both for easy and hard tasks, which is illustrated in Fig. A22. We speculate this is due to the increase in the number of trainable parameters: with PEFT methods we train just a few percent of all the parameters depending on the method and classifier head size, whereas with full fine-tune we train *all* the parameters ($50\times$ to $200\times$ more parameters), leading to an increase in average gradient size. Fig. A21 depicts to situation in terms of relative retained weights. Even in the hardest case (ViT-Tiny, $\varepsilon=0.5$) the gradients of the most difficult classes are not over-clipped with the smallest (useful) clipping bound.

Fig. A22 shows the full grid results, demonstrating that both the easy and hard classes benefit from a large clipping bound. As the amount of noise increases (smaller $\varepsilon$) the model starts to prefer a slightly smaller (though still large) clipping bound. We explain this in Figs. A23 to A25 which show the gradient norm distributions for three clipping bounds picked from the grid from a reasonable range around the optimum. In all cases most of the gradient norms are way above the typical clipping

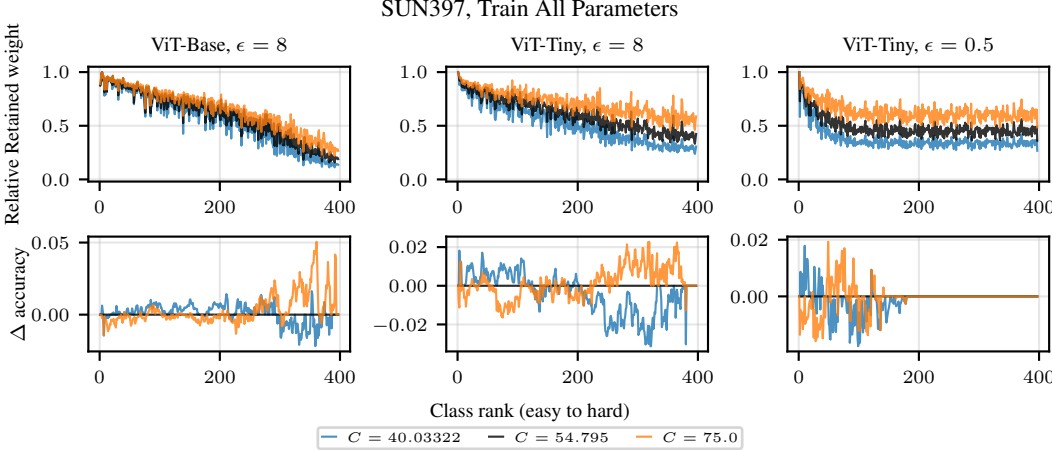

Figure A21: Per-class effects of gradient clipping under increasing task difficulty (left to right), SUN397, 8 epochs; $\delta=10^{-5}$. Class rank is based on sorting classes according to their per-class accuracies. Top: relative retained weight after clipping, computed per class and normalized to the class with the largest retained weight in the baseline ($C$=54.795). The effective clipping bounds are large enough—due to the increase in the average gradient size—to keep even the hardest classes gradient signal alive.

bounds used (e.g. 0.1, 1.0) and even above the clipping bounds that we have used in this paper for the hard cases (e.g. 10, 26, 42). We speculate that with smaller models (fewer trainable parameters), the easy-vs-hard effect will reappear, as hinted in our from-scratch experiment with WRN-16-4 (see Appendix I.1).

Lastly, we note that training large models with full fine-tune is costly and typically leads at it bets to on-par accuracy with PEFT methods Tobaben et al. (2023) or even degraded accuracy compared, especially under tight privacy Ke et al. (2024); Tobaben et al. (2023).

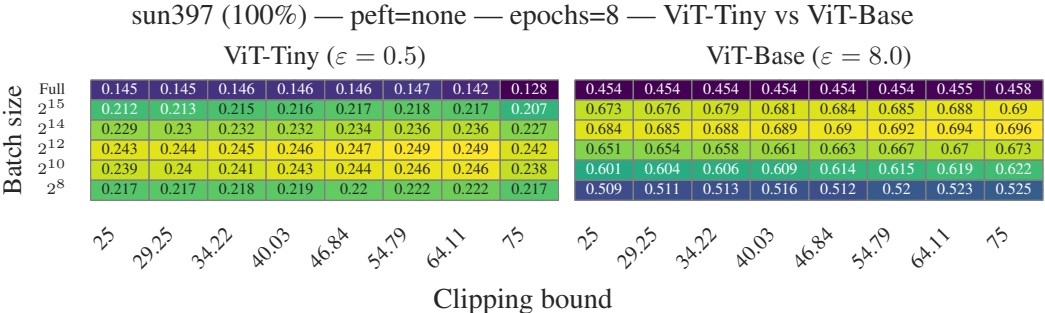

Figure A22: Comparison of the accuracies achieved over $(B, C)$ grid with hard (ViT-Tiny, $\varepsilon$=0.5) and easy (ViT-Hard, $\varepsilon$=8) tasks while fine-tuning all the parameters, SUN397, 8 epochs; $\delta=10^{-5}$. Learning rate is tuned separately for each configuration. Due to the increase in average gradient size, the optimal clipping bounds are much larger than with PEFT fine-tuning. With full fine-tune hard tasks would benefit from even larger clipping bounds, but these start to deteriorate learning due to the increase in injected noise.

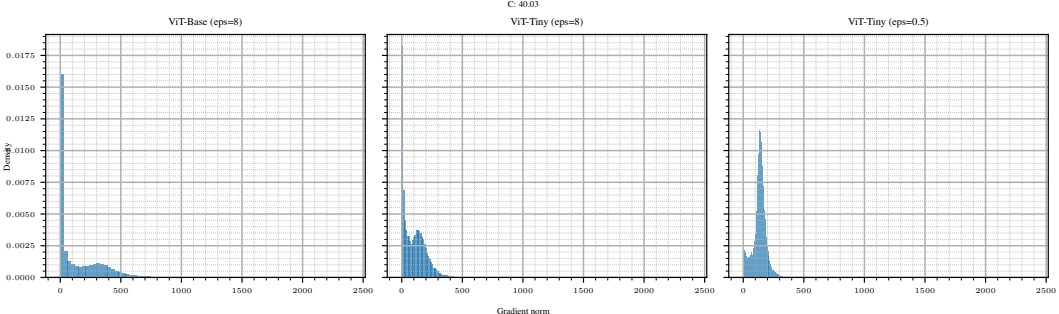

Figure A23: Histogram of gradient norms in easy (ViT-Base, $\varepsilon$=8), medium (ViT-Tiny, $\varepsilon = 8$), and hard (ViT-Tiny, $\varepsilon$=0.5) tasks, $C$=40.03; SUN397, 8 epochs; $\delta$=$10^{-5}$. The average gradient norm distribution has mass way above the usual clipping bounds.

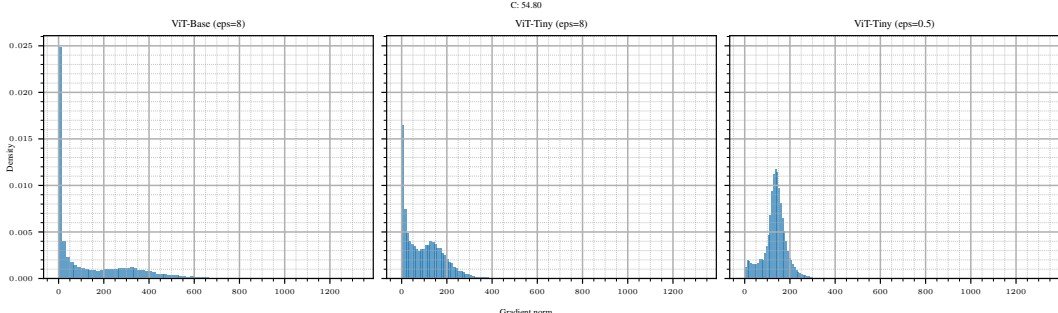

Figure A24: Histogram of gradient norms in easy (ViT-Base, $\varepsilon$=8), medium (ViT-Tiny, $\varepsilon = 8$), and hard (ViT-Tiny, $\varepsilon$=0.5) tasks, $C$=54.8; SUN397, 8 epochs; $\delta$=$10^{-5}$. The average gradient norm distribution has mass way above the usual clipping bounds.

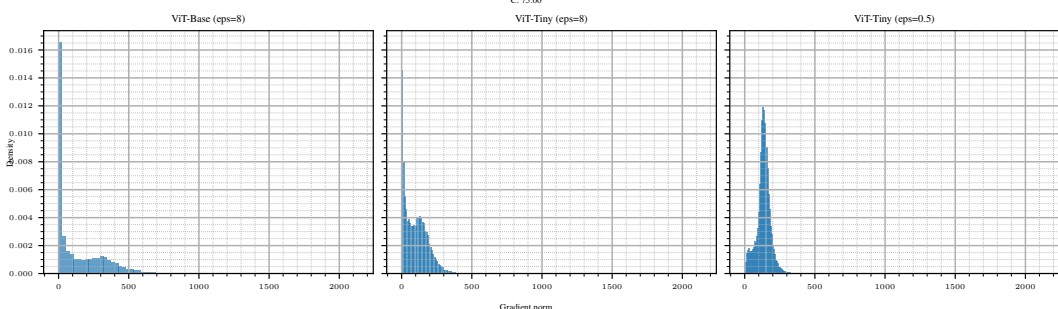

Figure A25: Histogram of gradient norms in easy (ViT-Base, $\varepsilon$=8), medium (ViT-Tiny, $\varepsilon = 8$), and hard (ViT-Tiny, $\varepsilon$=0.5) tasks, $C$=75; SUN397, 8 epochs; $\delta$=$10^{-5}$. The average gradient norm distribution has mass way above the usual clipping bounds.

## J  GRADIENT NORM DISTRIBUTIONS DURING TRAINING FOR SMALL AND LARGE $C$

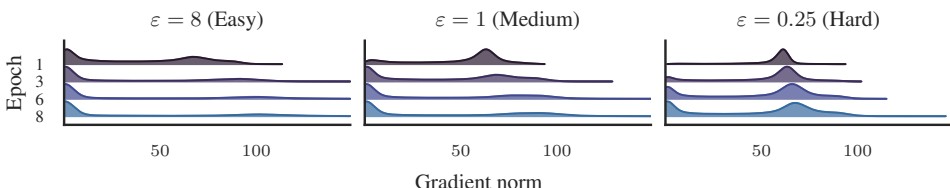

Figure A26: Gradient norm distributions over training epochs for ViT-Tiny on SUN397 (8 epochs; $\delta=10^{-5}$, $C=0.1$). Learning-problem difficulty increases from left to right as $\varepsilon$ decreases. Thicker regions imply higher probability mass. As difficulty increases, the distributions shift toward higher gradient norms.

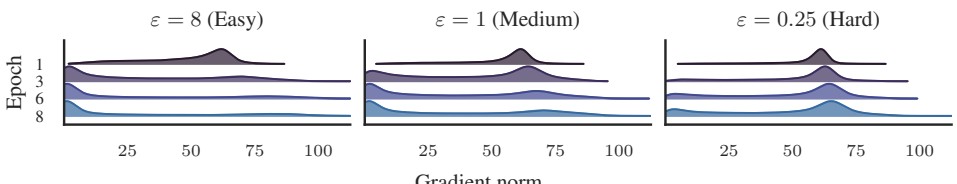

Figure A27: Gradient norm distributions over training epochs for ViT-Tiny on SUN397 (8 epochs; $\delta=10^{-5}$, $C=42$). Learning-problem difficulty increases from left to right as $\varepsilon$ decreases. Thicker regions imply higher probability mass. As difficulty increases, the distributions shift toward higher gradient norms.

# K   PER-CLASS EFFECTS OF GRADIENT CLIPPING

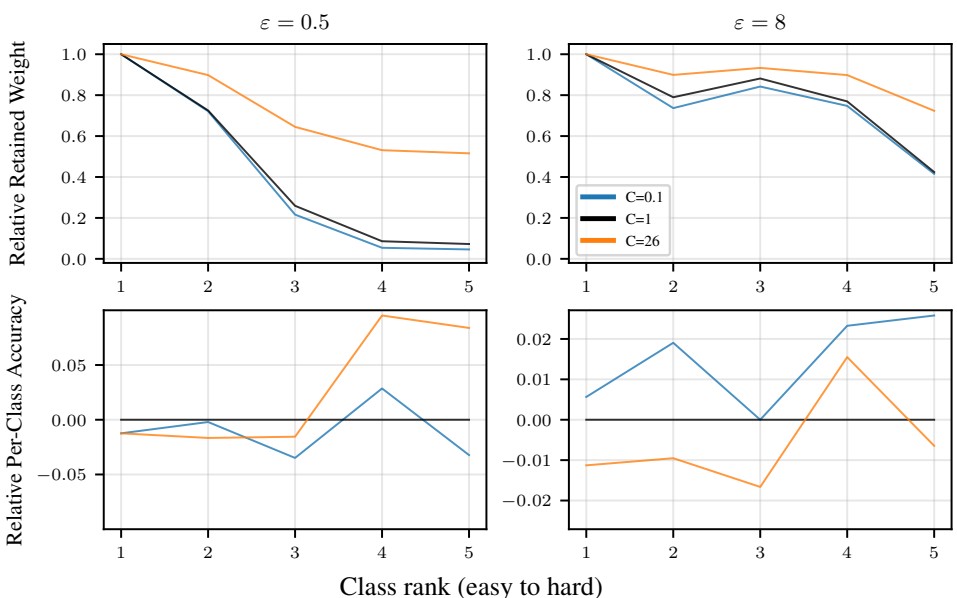

Figure A28: Per-class effects of gradient clipping under increasing task difficulty (left to right). Trained on Cassava with 8 epochs and tuned hyperparameters; $\delta = 10^{-5}$.

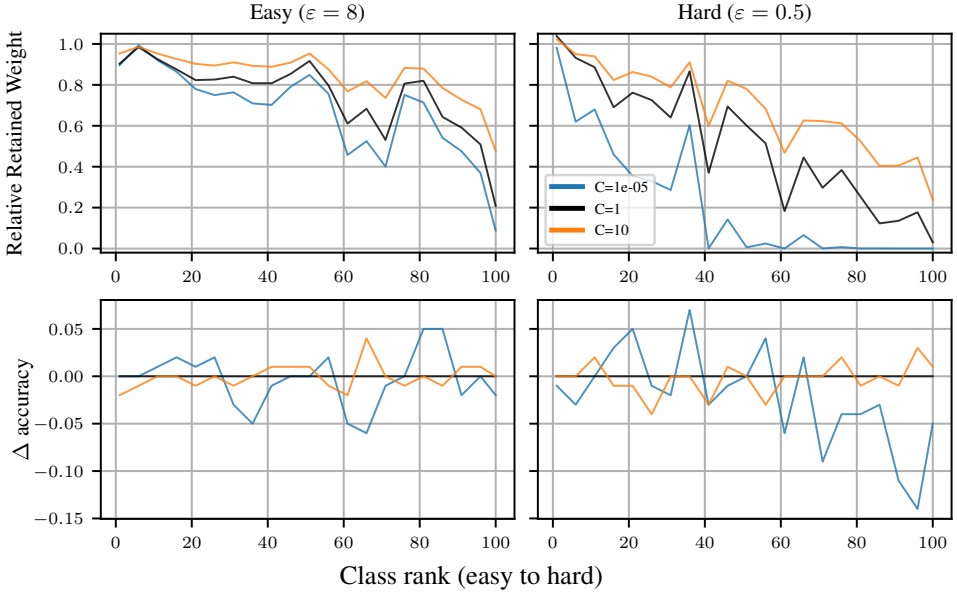

Figure A29: Per-class effects of gradient clipping under increasing task difficulty (left to right). Trained on CIFAR-100 with 8 epochs and tuned hyperparameters; $\delta = 10^{-5}$.

## L    MEAN GRADIENT NORM OVER ITERATIONS

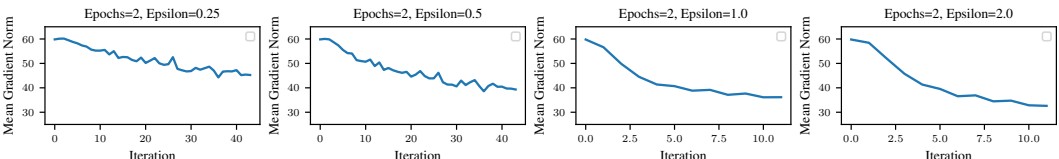

Figure A30: The average norm of gradient trace on SUN397 with ViT-Tiny, 2 epochs

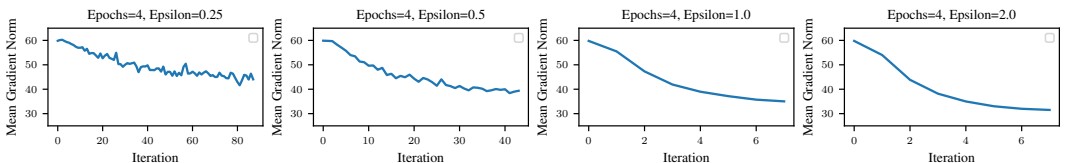

Figure A31: The average norm of gradient trace on SUN397 with ViT-Tiny, 4 epochs

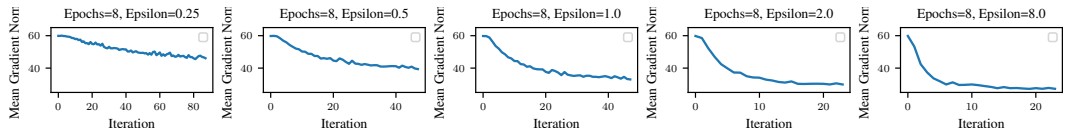

Figure A32: The average norm of gradient trace on SUN397 with ViT-Tiny, 8 epochs

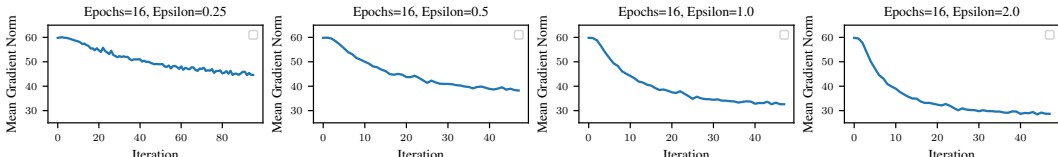

Figure A33: The average norm of gradient trace on SUN397 with ViT-Tiny, 16 epochs

## M    CUMULATIVE NOISE VS. EFFECTIVE NOISE

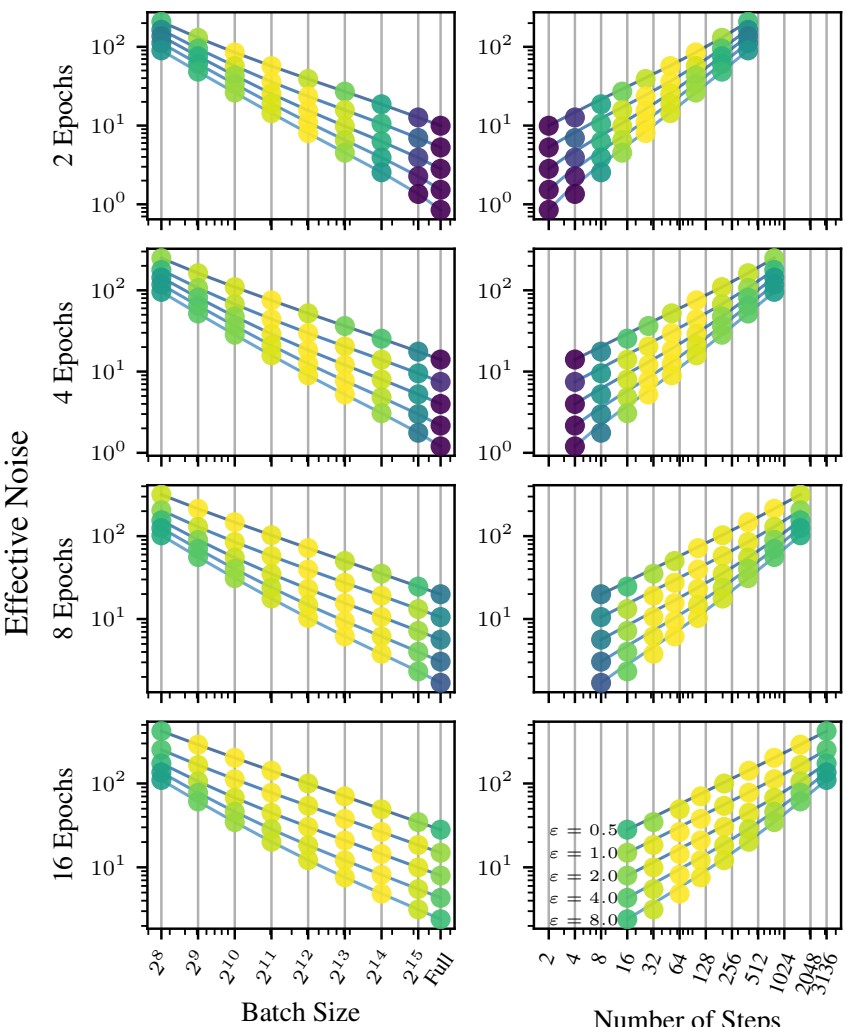

Figure A34: Effective noise, batch size and the number of steps under fixed-epoch DP-Adam on CIFAR-100 (ViT-Tiny, FiLM; $N{=}50000$, $\delta{=}10^{-5}$). Results are averaged over 3 seeds. The learning rate and clipping bound are tuned separately for each point.

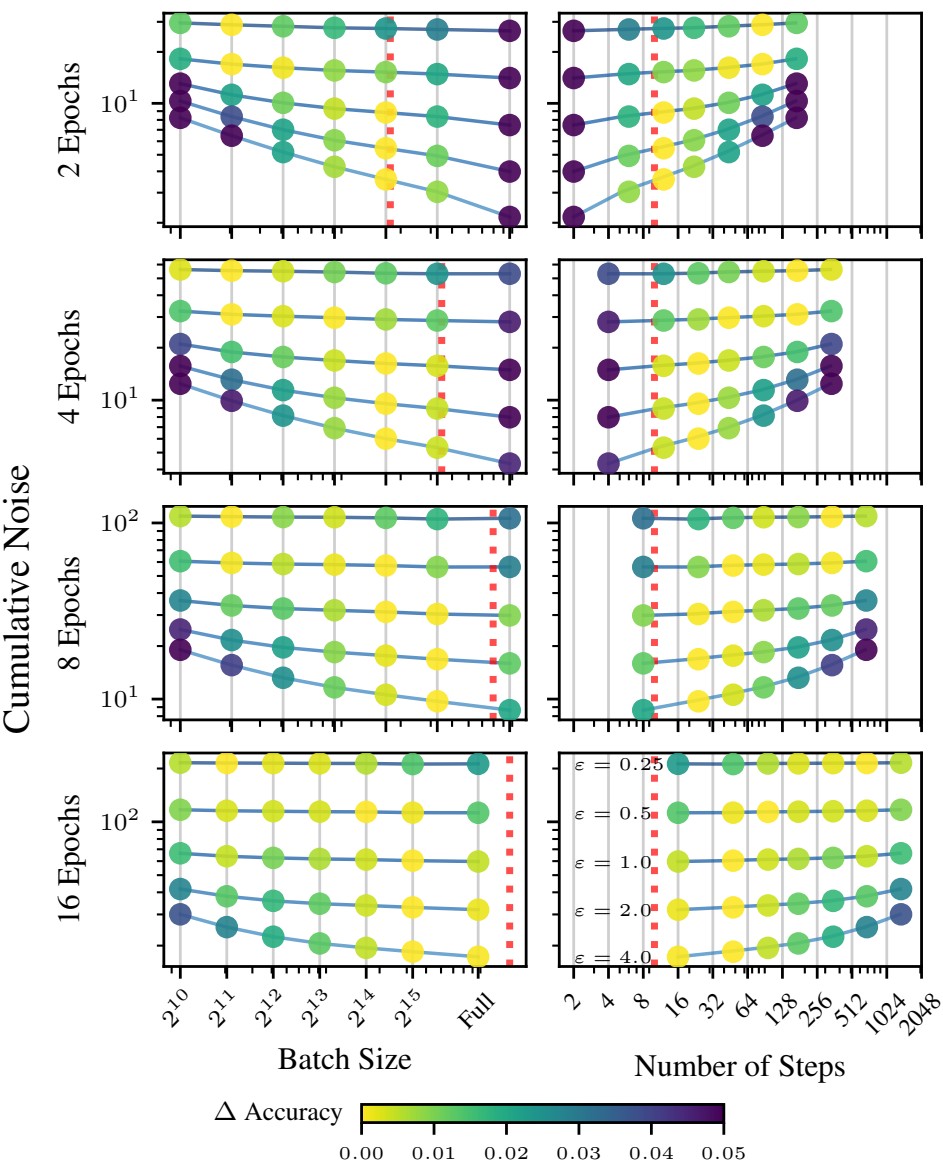

Figure A35: Cumulative noise, batch size and the number of steps under fixed-epoch DP-Adam on SUN397 (ViT-Tiny, FiLM; $N$=87002, $\delta$=$10^{-5}$). Results are averaged over 3 seeds. The learning rate and clipping bound are tuned separately for each point.

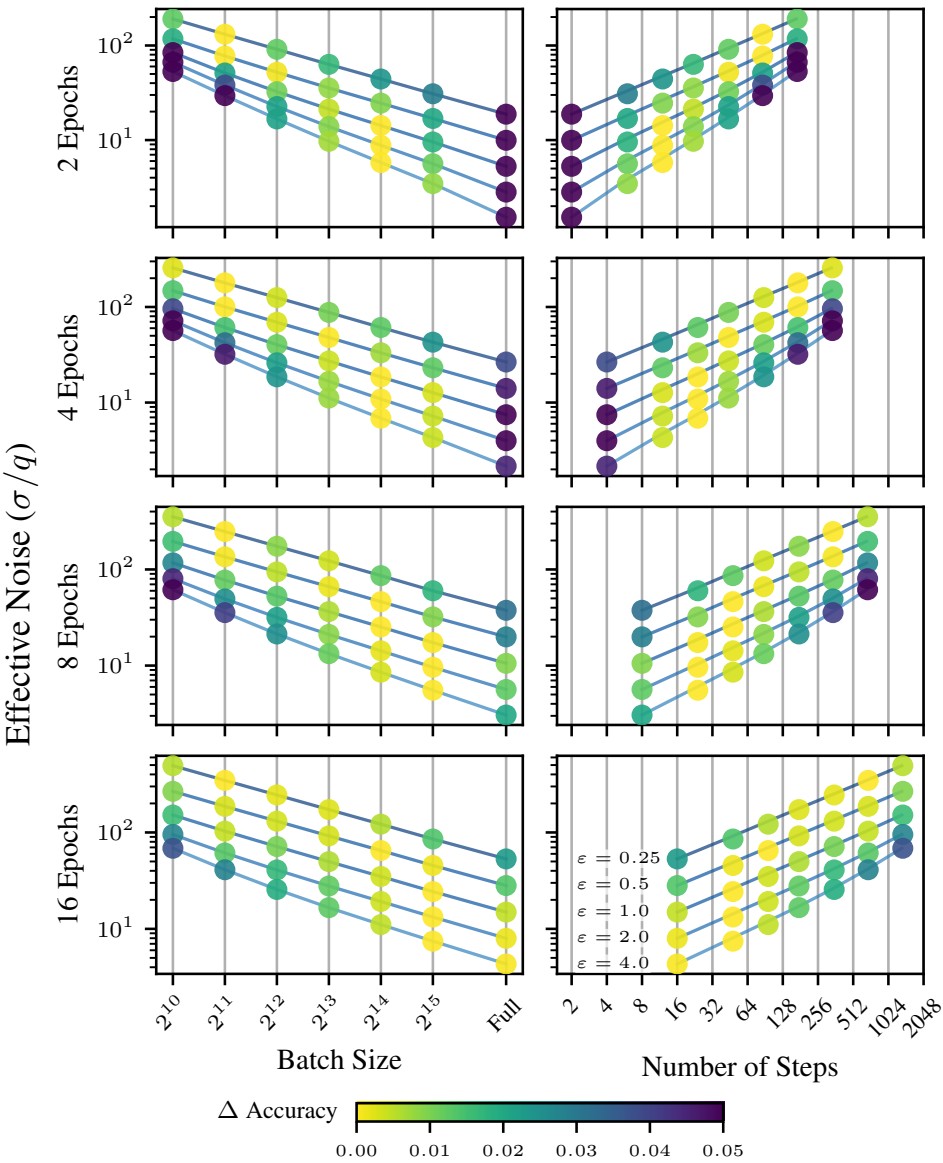

Figure A36: Effective noise, batch size and the number of steps under fixed-epoch DP-Adam on SUN397 (ViT-Tiny, FiLM; $N=87002$, $\delta=10^{-5}$). Results are averaged over 3 seeds. The learning rate and clipping bound are tuned separately for each point.

