# OpenReview forum: "On Optimal Hyperparameters for Differentially Private Deep Transfer Learning"
_ICLR.cc/2026/Conference — ICLR 2026 Poster_

### Official Review · Reviewer_GYoL · 2025-10-27

**Soundness:** 3
**Presentation:** 3
**Contribution:** 4
**Rating:** 6
**Confidence:** 4

**Summary:**

This paper focuses on clipping threshold and batch size. The paper studies how these two usually-fixed hyperparameters in differentially private (DP) transfer learning should depend on privacy level, compute budget, and backbone strength.

The authors (i) document that empirical optima contradict a common theoretical takeaway ("tighter privacy $\Longrightarrow$ smaller C"), (ii) provide an analysis that explains when larger C helps under tight privacy via a new MSE-based optimal clipping result (Theorem 5.1), and (iii) argue that in fixed‑epoch (compute‑bounded) training, cumulative DP noise ​$\sigma\sqrt{T}$ together with a minimum-steps constraint better explains optimal batch size than per‑step noise heuristics developed for fixed‑steps regimes. The empirical study uses DP‑Adam with PRV accounting ($\delta$= 1e−5) and extensive grid search over LR, B, C on SUN397, Cassava, CIFAR‑100 (and a 10% subset), comparing ViT‑Base vs ViT‑Tiny (and some ResNet‑50 in the appendix).

For the batch size part, the authors show that “effective noise” plots used in fixed‑steps heuristics (e.g., $\sigma/q$ ) monotonically favor full batch under fixed‑epochs, hence are uninformative (Figure 6). Instead, plotting cumulative noise ​$\sigma\sqrt{T}$ versus B reveals plateaus under tight privacy where moderate B can match or beat large B, provided a minimum number of steps is met (Figure 7). The appendices give dataset‑specific grids (Table A1) and many robustness plots across $\epsilon$'s and epochs.

**Strengths:**

1. Many DP transfer‑learning pipelines fix C and B across tasks; this paper convincingly shows that practice leaves accuracy on the table and can be especially harmful in hard regimes (e.g., small privacy budget, limited compute, weaker backbones)

2. Theorem 5.1 explains when the optimum moves the other way because the gradient distribution itself shifts under DP. Figure 3 and the Figure 4 make this intuitive and actionable.

3. The fixed‑epoch framing matches many real‑world budgets. The "cumulative noise plateau" and "minimum steps" interpretation is intuitive and well‑visualized.

4. Implementation details like hyper-parameter grids, (un)normalized-DP version, and class imbalance conditions are well documented and explained.

**Weaknesses:**

**Major concerns**
1. Previous works have shown that the choice of fine-tuning methods and initialization of heads could influence the DP performance, e.g. [1] [2]. However, all main experiments in this paper fine‑tune with FiLM (scale/bias of normalization layers + head), leaving about 0.5–1.5% of parameters trainable (Section Fine‑tuning). While the authors vary backbones (ViT‑Tiny/ViT‑Base and some ResNet‑50), the parameterization is fixed; Table A1 confirms all grids are under FiLM. In the discussion they state the mechanisms are “general” and expect transfer to full‑model tuning, but this is not demonstrated. This design choice narrows the external validity of the paper’s conclusions about optimal clipping C and batch size B.

References:

[1] Want et al. 2024. Neural Collapse meets Differential Privacy: Curious behaviors of NoisyGD with Near-Perfect Representation Learning. ICML.

[2] Ke et al. 2024. Characterizing the Training Dynamics of Private Fine-tuning with Langevin diffusion. NeurIPS Workshop on Fine-Tuning in Modern Machine Learning: Principles and Scalability.

**General weakness**

1. Most results are with DP‑Adam on image classification and parameter‑efficient fine‑tuning. While some ResNet‑50 and dataset diversity exist in the appendix, it remains unclear how fully the conclusions transfer to DP‑SGD, full‑model fine‑tuning, or other modalities (NLP, speech). The paper argues mechanisms are general, but evidence is image‑centric.

2. The "minimum steps + cumulative noise plateau" guidance is compelling visually, but a practitioner‑ready recipe (how to pick/estimate the steps threshold per dataset/task without extra privacy spend) is not spelled out. For example, in Figure 7 the dashed 20‑step threshold is illustrative, and Appendix A notes dataset‑dependence, but there is no principled estimator.

3. The study performs extensive grid searches, tuning LR jointly with C,B per configuration, which is appropriate for understanding behavior but raises the usual question of privacy leakage from tuning in practical deployments. The paper cites work on DP hyperparameter tuning, but it does not quantify or discuss the privacy cost of its own search.

**Theorem 5.1 is confusing to me for several reasons.**

- Weak theory–practice bridge. The current MSE criterion is explanatory rather than prescriptive. Without a descent‑type link (the small lemma suggested above) or a correlation study tying MSE proxies to accuracy across your sweeps (Fig. 2 and Fig. 4), the reader is asked to accept that “lower MSE $\approx$ better” on faith. Clarifying this would substantially strengthen the contribution.

- State prominently that $N_C$ and $G_C$ are used only for analysis in this paper, and provide one of the DP‑compliant procedures above for any future adaptive use. As written, a well‑meaning practitioner could implement a non‑private adaptation.

- Recasting Theorem 5.1 as a piecewise‑explicit optimizer (with convexity and boundary cases spelled out) would eliminate the fixed‑point ambiguity and address the "hand‑wavy" perception. Please centralize all assumptions next to the theorem.

**Some other minor issues.**
- In Figure 2, the plots only show relative accuracy differences. It would always be helpful to show the absolute accuracy value of each experiment result here. Readers would always prefer complete information rather than a visually straightforward but incomplete figure. And it would be helpful to provide a zero-shot baseline (i.e. no fine-tuning at all). The zero-shot baseline would provide a basic sanity check when the authors argue that the number of parameters is positively correlated to the "model capacity".

**Questions:**

- In Figure 4, could you explain more on how you determine which classes are harder than others?

- Could you turn Theorem 5.1 into a procedure? In other words, can you propose a privacy‑respecting estimator for the statistics that define the optimal $C$? How sensitive would such a method be to noise?

- How would you incorporate DP hyperparameter tuning (e.g., subsampled validation with DP bandit/BO) into your workflow to preserve the reported gains when tuning itself must be private?

---

> ### Author Response · Authors · 2025-12-03
>
> We thank the reviewer for carefully reading our paper and raising several important points, which we will address next! We will start with the weaknesses pointed out.
>
> > In the discussion they state the mechanisms are “general” and expect transfer to full‑model tuning, but this is not demonstrated. This design choice narrows the external validity of the paper’s conclusions about optimal clipping C and batch size B.
>
> We have now added an experiment with full fine-tuning (see Appendix I.3), where we find that--especially with large models--the increased count in trainable parameters results in much larger average gradient norm, preventing the use of small clipping bounds for easy tasks. Under this setting, both the easy and hard task prefer very large clipping bounds. We have also updated the discussion in Section 7 to reflect these findings.
>
> > 1. Most results are with DP‑Adam on image classification and parameter‑efficient fine‑tuning. While some ResNet‑50 and dataset diversity exist in the appendix, it remains unclear how fully the conclusions transfer to DP‑SGD, full‑model fine‑tuning, or other modalities (NLP, speech). The paper argues mechanisms are general, but evidence is image‑centric.
>
>
> We have now added a text classification experiment: fine-tuning the 20 Newsgroups dataset using DistilBERT with LoRA adapters. We observe milder, but similar trends and show with the retained weight analysis that the underlying mechanism matches our analysis. The results can be found in Appendix F.2.
>
> Furthermore, in Appendix F.1 we repeat the main SUN397 image classification experiment with LoRA. We note that in this task LoRA seems to be remarkably stable in the easy case, but in the hard case the clipping bound behavior matches our analysis. Again, we demonstrate using the retained weight analysis that the underlying mechanism seems to be the same.
>
> Lastly, we added an experiment with full fine-tuning (see Appendix I.3), where we find that--especially with large models--the increased count in trainable parameters results in much larger average gradient norm, preventing the use of small clipping bounds for easy tasks.
> Under this setting, both the easy and hard task prefer very large clipping bounds. Linear probing we had to unfortunately to omit, due to the workload.
>
> > 2. The "minimum steps + cumulative noise plateau" guidance is compelling visually, but a practitioner‑ready recipe (how to pick/estimate the steps threshold per dataset/task without extra privacy spend) is not spelled out. For example, in Figure 7 the dashed 20‑step threshold is illustrative, and Appendix A notes dataset‑dependence, but there is no principled estimator.
>
> We agree that a practitioner-ready recipe for estimating the step threshold without additional privacy spend would be extremely valuable. Our current paper focuses on characterizing the mechanisms behind the minimum-steps and cumulative-noise effects, and a full recipe for automatic threshold selection was beyond the scope of this work. We view our analysis as a foundation for developing such methods. One particularly promising direction, is to quantify the learning-problem difficulty and adjust the step threshold and other hyperparameters as a function of this quantity.
>
> > 3. The study performs extensive grid searches, tuning LR jointly with C,B per configuration, which is appropriate for understanding behavior but raises the usual question of privacy leakage from tuning in practical deployments. The paper cites work on DP hyperparameter tuning, but it does not quantify or discuss the privacy cost of its own search.
>
> However, we would like to point the reviewer to prior work [1], which provides empirical evidence that performing hyperparameter optimization using the training data does not lead to increased membership-inference vulnerability in DP deep transfer learning. Moreover, if one wishes to perform hyperparameter tuning under DP, it is possible to do so privately using existing DP-HPO methods [2,3], for example by randomly sampling hyperparameters from a predefined set.
>
> [1] Pradhan et al. 2025. Hyperparameters in Score-Based Membership Inference Attacks, SaTML.
>
> [2] Liu et al. 2019. Private selection from private candidates, SIGACT.
>
> [3] Papernot et al. 2022. Hyperparameter Tuning with Renyi Differential Privacy, ICLR.

---

> ### Author Response · Authors · 2025-12-03
>
> Next, we will address the points about our theory.
>
> > 1. Weak theory–practice bridge. The current MSE criterion is explanatory rather than prescriptive. Without a descent‑type link (the small lemma suggested above) or a correlation study tying MSE proxies to accuracy across your sweeps (Fig. 2 and Fig. 4), the reader is asked to accept that ``lower MSE $\approx$ better'' on faith. Clarifying this would substantially strengthen the contribution.
>
> As requested, we have now added Theorem 5.4 and Corollary 5.5 to directly connect the per-step MSE minimization to gradient descent rates (for smooth functions): we show that minimizing the per-step MSE directly improves a per-step bound on loss improvement, recovering the standard non-DP result when MSE is 0.
>
> > 2. State prominently that $N_c$ and $G_c$ are used only for analysis in this paper, and provide one of the DP‑compliant procedures above for any future adaptive use. As written, a well‑meaning practitioner could implement a non‑private adaptation.
>
> We now clearly and explicitly state this immediately after Thm 5.2 (previously Thm 5.1).
>
> > 3. Recasting Theorem 5.1 as a piecewise‑explicit optimizer (with convexity and boundary cases spelled out) would eliminate the fixed‑point ambiguity and address the "hand‑wavy" perception. Please centralize all assumptions next to the theorem.
>
> We have revised Thm 5.2 (previously Thm 5.1) as proposed, including the boundary cases, and included the necessary assumptions as a statement next to every theorem (Assumption 5.1, 5.3).
>
> Next, we will comment on the other minor issues raised by the reviewer:
>
> > 1. In Figure 2, the plots only show relative accuracy differences. It would always be helpful to show the absolute accuracy value of each experiment result here. Readers would always prefer complete information rather than a visually straightforward but incomplete figure.
>
> Unfortunately, due to workload, we weren't able to produce the absolute accuracy figures. However, we have now included those for the additional experiments we have done. They are available in the heatmaps in Appendix, e.g., Figure A12.
>
> > And it would be helpful to provide a zero-shot baseline (i.e. no fine-tuning at all). The zero-shot baseline would provide a basic sanity check when the authors argue that the number of parameters is positively correlated to the "model capacity".
>
> We are not quite understanding what the reviewer is asking for; in a zero-shot baseline, the batch size and clipping bound are meaningless, as the model has not been trained under DP at all.
>
> Lastly, we will address the questions raised by the reviewer.
>
> > 1. In Figure 4, could you explain more on how you determine which classes are harder than others?
>
> We evaluate the per-class accuracies and determine the difficulty based on that. We have now clarified this in the Figure 4 caption.
>
> > 2. Could you turn Theorem 5.1 into a procedure? In other words, can you propose a privacy‑respecting estimator for the statistics that define the optimal ? How sensitive would such a method be to noise?
>
> This is an interesting proposal, but not immediately doable, as privatizing all the quantities in the Thm will add plenty of noise to the results. We must therefore leave the exploration of such a method for future work.
>
> > 3. How would you incorporate DP hyperparameter tuning (e.g., subsampled validation with DP bandit/BO) into your workflow to preserve the reported gains when tuning itself must be private?
>
> As mentioned earlier, our work is not intended to serve as a workflow for selecting hyperparameters. The 3-dimensional grid search is used purely to analyze how the optimal hyperparameters behave, not as a practical HPO method. The goal of our work is to understand the underlying mechanisms that cause this differences, rather than to propose an algorithm for (private) HPO.

---

### Official Review · Reviewer_yubP · 2025-10-30

**Soundness:** 3
**Presentation:** 3
**Contribution:** 3
**Rating:** 6
**Confidence:** 3

**Summary:**

This paper investigates the optimization strategies for hyper-parameters in DP finetuning, including the clipping bound $C$ and the batch size $B$. For the clipping bound $C$, the author presents a finding that under strict privacy budget, a larger clipping bound leads to improved model performance. For the batch size $B$, the paper proposes a principle for selecting the optimal $B$ on the fixed-epochs setting. This rule aims to overcome the limitations of traditional fixed-step theories by maximizing the number of optimization steps $T$ (via minimizing $B$) while ensuring that the cumulative DP noise ($\sigma \sqrt{T}$) remains within its near-optimal plateau region.

**Strengths:**

1. The paper demonstrates that the clipping bound $C$ and batch size $B$ should be dynamically adjusted based on the difficulty of the learning task, which is different from the traditional approach with fixed hyper-parameters.
2. The paper introduces the perspective of "gradient re-weighting," which provides a detailed analysis of how the clipping bound $C$ affects the learning process.
3. The paper highlights the necessity of jointly tuning $C$, $B$, and the learning rate, noting their complex interactions.
4. Comprehensive experiments are conducted.

**Weaknesses:**

1. The study relies only on the Vision Transformer; it would be beneficial to explore various models to verify the effect of pre-trained model capability on optimal clipping. At a minimum, some discussion of future work is necessary.

2. It would be great if the authors could discuss other related works concerning optimal hyperparameters in DP fine-tuning, especially the learning rate. For instance, [A] pointed out that if the features of a ViT are extracted near-perfectly, then DP fine-tuning is less sensitive to hyperparameters. This implies that in some extreme cases with near-perfect features, optimal hyperparameter tuning may not be necessary—a conclusion that differs from this paper, which considers cases where pre-training extracts near-perfect features for the fine-tuning datasets. However, [B] shows that if the features are not extracted perfectly, then optimal hyperparameters, especially an optimal learning rate, are necessary; otherwise, the features may be degraded by the noise, a conclusion that aligns with this work.


[A] Neural Collapse Meets Differential Privacy: Curious Behaviors of NoisyGD with Near-perfect Representation Learning. Wang et al., ICML, 2024.

[B] Why Does Private Fine-Tuning Resist Differential Privacy Noise? A Representation Learning Perspective. Zhao et al., ICLR 2025 Data-FM Workshop, 2025.

3.  The term "learning problem difficulty" is central to the paper but seems to be used informally (please correct me if I have missed a definition). It would be beneficial to define or characterize it more formally, perhaps as a function of the privacy budget (or noise level), model capability, the size of datasets. A brief discussion in Section 5 would suffice.

**Questions:**

See the weaknesses part.

---

> ### Author Response · Authors · 2025-12-03
>
> We would like to thank the reviewer for the trouble of carefully reading through our manuscript and helping us to make our paper better!
>
> > 1. The study relies only on the Vision Transformer; it would be beneficial to explore various models to verify the effect of pre-trained model capability on optimal clipping. At a minimum, some discussion of future work is necessary.
>
> We did also evaluate our findings on ResNet-50 and observed the same trend. For clarity, we now explicitly mention the ResNet-50 result in the first paragraph of Section 5 where we point to additional results in the appendix. For convenience, the plot can be found in Appendix E.2, Figure A6(a).
>
> > 2. It would be great if the authors could discuss other related works concerning optimal hyperparameters in DP fine-tuning, especially the learning rate. For instance, [A] pointed out that if the features of a ViT are extracted near-perfectly, then DP fine-tuning is less sensitive to hyperparameters. This implies that in some extreme cases with near-perfect features, optimal hyperparameter tuning may not be necessary—a conclusion that differs from this paper, which considers cases where pre-training extracts near-perfect features for the fine-tuning datasets. However, [B] shows that if the features are not extracted perfectly, then optimal hyperparameters, especially an optimal learning rate, are necessary; otherwise, the features may be degraded by the noise, a conclusion that aligns with this work.
>
> We have now cited the referenced papers and included a brief discussion of the related work about feature quality and the robustness of learning at the end of Section 5.2.
>
> > 3. The term "learning problem difficulty" is central to the paper but seems to be used informally (please correct me if I have missed a definition). It would be beneficial to define or characterize it more formally, perhaps as a function of the privacy budget (or noise level), model capability, the size of datasets. A brief discussion in Section 5 would suffice.
>
> We now explicitly define what we mean by the learning problem difficulty and added this to the introduction: "... learning–problem difficulty--that is mainly governed by the privacy budget, available data and compute, dataset and transfer complexity, and the pretrained backbone capability, which reflects both model complexity and pretraining quality." We thank the reviewer for pointing this out!

---

### Official Review · Reviewer_frYa · 2025-10-30

**Soundness:** 3
**Presentation:** 2
**Contribution:** 2
**Rating:** 2
**Confidence:** 4

**Summary:**

This paper investigates a mismatch between theoretical results and empirical findings regarding the optimal clipping threshold C in relation to the privacy budget $\varepsilon$. Thus, for DP transfer learning, the paper proposes using a gradient re-weighting mechanism to justify using a larger C with a smaller $\varepsilon$, and analyzing the total accumulated DP noise to inform batch size selection.

**Strengths:**

- The paper investigates an interesting idea: that previous approaches using a small, fixed C (e.g., 0.1 or 1) might be suboptimal.
- The authors provide a basic analysis of how DP noise affects the optimal clipping bound.
- Based on their observations, the authors’ proposed tuning approach performs better than using static hyperparameters.

**Weaknesses:**

Please refer to the Questions section.

**Questions:**

- The reviewer believes the gradient norm analysis needs to be further developed to fully support the paper's hypothesis. In Figure 3, the authors show the gradient norms after training with various $\varepsilon$ and C, but the gradient norm also varies during training. A discussion of this dynamic behavior seems to be missing.
-  Furthermore, the claim that the gradient norm gets bigger when not converged needs more support. In standard (and possibly DP) training, the gradient norm is often observed to increase towards the end of training (see [1], [2]), which seems to contradict the paper's explanation (i.e., that large norms are simply due to non-convergence). Please provide more backing for this claim.

    [1] Why Gradients Rapidly Increase Near the End of Training, 2025.

    [2] Penalizing Gradient Norm for Efficiently Improving Generalization in Deep Learning, ICML 2022.

-  Is there any significant difference between analyzing 'from scratch' and 'transfer learning' in terms of your analysis or empirical findings? Would your conclusions hold for 'from scratch' training?
-  Can you test your proposed framework on other SOTA DP training methods, such as those mentioned in the related works?

---

> ### Author Response · Authors · 2025-11-12
>
> We thank the reviewer for the valuable feedback, it is greatly appreciated!
>
> Before addressing the rebuttal in full, we would request a quick clarification.
>
> > Can you test your proposed framework on other SOTA DP training methods, such as those mentioned in the related works?
>
> Could you kindly clarify to which SOTA DP method you would like for us to compare against?

---

> ### Comment · Reviewer_GYoL · 2025-11-12
>
> DP-LoRA could be a reasonable choice if the authors have limited time and computation resources.

---

> ### Comment · Reviewer_frYa · 2025-11-13
>
> Thanks for the quick response from the authors and valuable comments from Reviewer GYoL.
>
> I do not say to compare "with" SOTA DP training, but test your framework on other DP training methods.
>
> I mean that the clipping bound is very important for private training, but it is also highly dependent on the settings.
>
> Thus, I think "augmentation multiplicity", as in [1], is a natural choice for training large models in a DP way these days.
>
> Or, as the reviewer GYoL commented, the authors can check the performance in a DP-LoRA setting.
>
> [1] Unlocking High-Accuracy Differentially Private Image Classification through Scale, DeepMind, 2022.

---

> ### Author Response · Authors · 2025-12-03
>
> We thank the reviewer for the comments! We will now address the questions in order.
>
> > 1. The reviewer believes the gradient norm analysis needs to be further developed to fully support the paper's hypothesis. In Figure 3, the authors show the gradient norms after training with various $\varepsilon$ and C, but the gradient norm also varies during training. A discussion of this dynamic behavior seems to be missing.
>
> To address the reviewer's point, we note that under tight privacy the training examples effectively become harder, then looking at the figures in Appendix K, it is apparent that hard examples (small $\varepsilon$ training) consistently have larger gradients compared to easy examples (large $\varepsilon$ training). And this is what our gradient reweighting view relies on: harder settings should maintain larger per-example gradients than easier settings.
>
> > 2. Furthermore, the claim that the gradient norm gets bigger when not converged needs more support. In standard (and possibly DP) training, the gradient norm is often observed to increase towards the end of training (see [1], [2]), which seems to contradict the paper's explanation (i.e., that large norms are simply due to non-convergence). Please provide more backing for this claim.
>
> Regarding [1], the phenomenon they describe is closely tied to learning-rate schedules and weight decay, neither of which we use; our learning rate is fixed, and we do not apply weight decay. Because of this, the mechanism responsible for late-stage gradient growth in [1] does not apply to our setup.
>
> For [2], the only explicit case in which the gradient norms grow toward the end of training is in their Figure 3 under the $\alpha=0.8$ regularization setting. In the remaining configurations in the same figure, the norms decrease as training progresses. Furthermore, since [2] introduces a sharpness-aware regularizer that substantially alters the optimization dynamics, we believe its behavior does not directly transfer to our fine-tuning setting.
>
> Nevertheless, we added a new experiment to empirically verify if the gradient norms indeed do grow toward the end of training. We plotted the mean gradient norm of various configurations as a function of iterations (gradient updates). In all cases the gradient norms shrink as training progresses supporting our original interpretation. These results can be found in Appendix K.
>
> > 3. Is there any significant difference between analyzing 'from scratch' and 'transfer learning' in terms of your analysis or empirical findings? Would your conclusions hold for 'from scratch' training?
>
> We also experimented with from-scratch training (see Appendix I.1) that indicate the trends reported in the fine-tuning setting also transfer to from-scratch setting. Figure A17 depicts this: hard classes suffer disproportionately from clipping, which matches our fine-tuning results. We find this holds especially when the parameter count of the model is not too large (see our full fine-tune experiments in Appendix I.3 for details).
>
> > 4. Can you test your proposed framework on other SOTA DP training methods, such as those mentioned in the related works?
>
> We have now included LoRA fine-tuning experiments (see Appendix F). Training with LoRA seems to be more stable in terms of clipping bound, especially in the "easy" case for the SUN397 image classification task. However, for the "hard" case, we can see performance degrading with smaller clipping bounds, while the easy ones perform great, as predicted by our analysis.
> Furthermore, we plotted the retained weight plots for the LoRA experiments and they show that the underlying mechanism matches our analysis. These results can be found in Appendix F.1. Furthermore, we also highlight that our findings also apply to text classification tasks, which we have now added (Appendix F.2).

---

### Official Review · Reviewer_J42k · 2025-10-31

**Soundness:** 3
**Presentation:** 2
**Contribution:** 2
**Rating:** 2
**Confidence:** 3

**Summary:**

This paper systematically studies how two critical hyperparameters, gradient clipping bound (C) and batch size (BS), affect performance in fine-tuning large pretrained models under DP constraints. Empirical and theoretical analysis showing that the optimal C depends on the privacy noise scale $\sigma$ and gradient distribution, which itself shifts with privacy level, model capacity, and dataset difficulty.
The paper finds that
1. Larger clipping bounds C can improve performance under stronger privacy (smaller $\varepsilon$) and for weaker pretrained backbones.
2. The optimal BS depends jointly on privacy and compute budget: under tight privacy, a smaller BS may perform better, while a moderate BS is optimal under limited compute.
3. Fixing (C, BS) across tasks degrades performance, especially for harder datasets and tighter privacy settings.

**Strengths:**

1. Clear motivation: Hyperparameter selection is an important and open question in DP training.
2. Novel theoretical contribution: Theorem 5.1 provides a principled explanation for observed discrepancies between prior theory and practice regarding the clipping bound.
3. Insightful conceptual framing: Interpreting clipping as gradient re-weighting is intuitive and helps explain asymmetric behavior between easy and hard examples.

**Weaknesses:**

1. Since the scope of this paper is DP fine-tuning, it would be complete if language classification tasks were considered. Additionally, this paper only considered fine-tuning the classification head plus the scale and bias of normalization layers. It would be complete and more convincing if more baselines were compared, such as LoRA, linear probing, and full fine-tuning, given that the latter are widely used fine-tuning methods.
2. The section "Clipping as Gradient Re-Weighting" is insightful, and it would be better if this section were written more mathematically rigorously rather than in storytelling.
3. Writing: It would be clearer if the authors could summarize and highlight all the implications for practical hyperparameter tuning in bullet points in the introduction/Section 5 & 6.

**Questions:**

1. Line 30-33 claim that usually only LR is tuned for each separate problem, while other hyperparameters (BS and C) are fixed, with reference made to De et al. 2022. Could the authors clarify how De et al. 2022 is an instance? According to my understanding, De et al. 2022 conducted careful hyperparameter tuning.
2. For the plots that present the change of optimal C/BS under different settings, how are the other hyperparameters (LR, BS/C, epoch) selected? For example, each experiment in Fig. 2 (left) might require its own LR and BS, and it would be helpful to describe how they are selected. Do all the runs in Fig. 2(left) use the same LR and BS?

Overall, I suggest augmenting the experiments, improving Section 5.3, and improving the writing to make the insights for hyperparameter tuning more executable. After the clarification questions and concerns are addressed, I would be happy to raise my score.

---

> ### Author Response · Authors · 2025-12-03
>
> We heartily thank the reviewer for their trouble and insightful comments!
>
> We will first address the weaknesses.
>
> > 1. Since the scope of this paper is DP fine-tuning, it would be complete if language classification tasks were considered. Additionally, this paper only considered fine-tuning the classification head plus the scale and bias of normalization layers. It would be complete and more convincing if more baselines were compared, such as LoRA, linear probing, and full fine-tuning, given that the latter are widely used fine-tuning methods.
>
> We have now added a text classification experiment: fine-tuning the 20 Newsgroups dataset with DistilBERT using LoRA adapters. We observe milder, but similar trends and show with the retained weight analysis that the underlying mechanism matches our analysis (see Appendix F.2).
>
> Next, we repeated the main SUN397 image classification experiment with LoRA in Appendix F.1. We note that in this task LoRA seems to be remarkably stable in the easy case, but in the hard case the clipping bound behavior matches our analysis. Again, we demonstrate using the retained weight analysis that the underlying mechanism seems to be the same.
>
> Lastly, we added a full fine-tune experiment, where we find that--especially with large models--the increased count in trainable parameters results in much larger average gradient norm, preventing the use of small clipping bounds for easy tasks. In this setting, both the easy and hard task prefer very large clipping bounds, we refer to Appendix I.3 for detailed discussion.
>
> Linear probing we had to unfortunately to omit, due to the workload.
>
> > 2. The section "Clipping as Gradient Re-Weighting" is insightful, and it would be better if this section were written more mathematically rigorously rather than in storytelling.
>
> We have now incorporated the gradient re-weighting explanation in full to the main text with more mathematical rigor (Equation 4, Section 5.3).
>
> > 3. Writing: It would be clearer if the authors could summarize and highlight all the implications for practical hyperparameter tuning in bullet points in the introduction/Section 5 and 6.
>
> We have now added Table 1 summarizing practical tuning implications in the end of introduction section.
>
> Next we will address the remaining questions.
>
> > 1. Line 30-33 claim that usually only LR is tuned for each separate problem, while other hyperparameters (BS and C) are fixed, with reference made to De et al. 2022. Could the authors clarify how De et al. 2022 is an instance? According to my understanding, De et al. 2022 conducted careful hyperparameter tuning.
>
> Our statement uses "and/or," meaning we do not claim that both BS and C are always fixed. In De et al. (2022), Appendix B.1 explicitly states: "Therefore, to reduce the cost of hyper-parameter tuning, for all of our experiments in this paper, we fix C = 1." Regarding batch size, the paper does not clearly document how it was tuned, but to the best of our understanding, it was not re-tuned for each $\varepsilon$.
>
> > 2. For the plots that present the change of optimal C/BS under different settings, how are the other hyperparameters (LR, BS/C, epoch) selected? For example, each experiment in Fig. 2 (left) might require its own LR and BS, and it would be helpful to describe how they are selected. Do all the runs in Fig. 2(left) use the same LR and BS?
>
> In all our results, LR, BS, and C are jointly tuned from the full grid (see Table A1 for details of the grid) for each experiment. The only parameter we fix is the number of epochs. We appreciate the reviewer pointing this out; we realized that this detail was missing from the caption of Fig. 2 and we now explicitly mention this in the caption.

---

### Author Response · Authors · 2025-12-03

We thank all the reviewers for their careful reading of our work and the constructive feedback given, we very much appreciate the chance given to improve our work. We have now revised our manuscript accordingly and summarize the main changes below.

**Improved theory and clarified gradient re-weighting.**

As requested, we added new Theorem 5.4 and Corollary 5.5 to directly connected the per-step MSE from Theorem 5.2 to optimization rates, showing that a reduction in the per-step MSE improves the bound on loss improvement. We also rewrote the main Theorem 5.2 (previously 5.1) using piecewise analysis for clarity. Furthermore, we now state the assumptions (Assumption 5.1, 5.3) needed for all theorems next to the theorems.

We improved the theoretical presentation of the "clipping as gradient re-weighting" view by integrating the definitions directly into the main text (equation (4) in Section 5.3).


**Expanded empirical evaluation settings**

We extended the experimental scope to better support our claims about the generality of our analysis. In addition to the original ViT-based FiLM fine-tuning, we now experiment with:

(i) LoRA fine-tuning on image classification task with SUN397 (Appendix F.1),

(ii) LoRA fine-tuning on text classification task with the 20 Newsgroups dataset using DistilBERT model with LoRA adapters (Appendix F.2),

(iii) DP-SGD on image classification (Appendix I.2),

(iv) training from scratch with WideResNet-16-4 (Appendix I.1), and

(v) full (all parameter) fine-tuning with ViT-Base and ViT-Tiny (Appendix I.3).

Across these settings, we observe the same trends, with the exception of full fine-tuning with large models, which always prefers a very large clipping bound, discussed in detail in Appendix I.3.


**Gradient-norm dynamics and comparison to prior work.**

In response to concerns about the dynamics of gradient norms and their relation to convergence, we added experiments that track the mean gradient norm over iterations for multiple configurations (Appendix K). In all our settings, the gradient norms consistently decrease as training progresses, supporting our interpretation that harder configurations maintain larger per-example gradients.

**Other minor changes.**

We now define "learning-problem difficulty" in the introduction

We added a table of practical implications (Table 1) at the end of the introduction.

We improved the main text and caption of Fig. 2 to make it clear that the learning rate, batch size, and the clipping bound are jointly tuned using the task-specific grids in Table A1; only the number of epochs is fixed. We further clarified in Fig. 4 caption that "hard" classes are determined by their per-class accuracies.

For the new experiments in the appendix, we also include absolute-accuracy heatmaps .

We included a short discussion of related work on the quality of pretrained features at the end of Section 5.2.

---

### Meta-Review · Area_Chair_7cnn · 2026-01-15

**Summary:**

All reviewers agree that the topic of study in this work is interesting and novel. There were concerns about the experiments and clarity and authors have adequately addressed those in the rebuttal.

**Reviewer Concerns:**

Reviewers (J42k, yubP, and  GYoL) had asked for experiments on language, with different training methods (DP-LORA) and different architectures (Resnet). Authors provide these experiments int the rebuttal.

Reviewers J42k and frYa asked for more experimental and mathematical rigor for paper's claim on gradient norm during training. Authors provide additional experiments and math to address this.

Reviewer GYoL asks for extension of the main theorem 5.1 to connect it better with practice. Authors provided a new theory to address this.

Reviewer GYoL asks about quantifying the privacy cost of hyperparameter tuning. Authors do not address this comment adequately.

**Reviewer Scores:**

I believe all reviewers may increase their score in light of the new experiments. Authors have done a good job addressing reviewers editorial and clarification comments as well. The outstanding concern seems to be the privacy cost of hyper-parameter tuning.

---

### Decision · Program_Chairs · 2026-01-26

Accept (Poster)